# Transcription factors underlying photoreceptor diversity

**Juan M Angueyra[1]\*, Vincent P Kunze[1], Laura K Patak[1], Hailey Kim[1], Katie Kindt[2], Wei Li[1]\***

[1]Unit of Retinal Neurophysiology, National Eye Institute, National Institutes of Health, Bethesda, United States; [2]Section on Sensory Cell Development and Function, National Institute on Deafness and Other Communication Disorders, National Institutes of Health, Bethesda, United States

**Abstract** During development, retinal progenitors navigate a complex landscape of fate decisions to generate the major cell classes necessary for proper vision. Transcriptional regulation is critical to generate diversity within these major cell classes. Here, we aim to provide the resources and techniques required to identify transcription factors necessary to generate and maintain diversity in photoreceptor subtypes, which are critical for vision. First, we generate a key resource: a high-quality and deep transcriptomic profile of each photoreceptor subtype in adult zebrafish. We make this resource openly accessible, easy to explore, and have integrated it with other currently available photoreceptor transcriptomic datasets. Second, using our transcriptomic profiles, we derive an in-depth map of expression of transcription factors in photoreceptors. Third, we use efficient CRISPR-Cas9 based mutagenesis to screen for null phenotypes in F0 larvae (F0 screening) as a fast, efficient, and versatile technique to assess the involvement of candidate transcription factors in the generation of photoreceptor subtypes. We first show that known phenotypes can be easily replicated using this method: loss of S cones in *foxq2* mutants and loss of rods in *nr2e3* mutants. We then identify novel functions for the transcription factor Tbx2, demonstrating that it plays distinct roles in controlling the generation of all photoreceptor subtypes within the retina. Our study provides a roadmap to discover additional factors involved in this process. Additionally, we explore four transcription factors of unknown function (Skor1a, Sall1a, Lrrfip1a, and Xbp1), and find no evidence for their involvement in the generation of photoreceptor subtypes. This dataset and screening method will be a valuable way to explore the genes involved in many other essential aspects of photoreceptor biology.

**\*For correspondence:**
angueyra@nih.gov (JMA);
liwei2@nei.nih.gov (WL)

**Competing interest:** The authors declare that no competing interests exist.

## Editor's evaluation

This manuscript offers a valuable transcriptomic data set of known types of adult zebrafish photoreceptors (rod and cones). The study identifies a large set of differentially expressed transcription factors, many of which still have an unidentified function in photoreceptors and offers to the scientific community an interactive plotter to compare the present data with recent and similar studies. Using CRISPR F0 screening, the study shows that the two tbx2 zebrafish paralogues are involved in photoreceptors specification beyond what is currently known. The study uses a solid methodology that could be applied to other retinal cell types or other tissues.

## Introduction

Creating cells with diverse features is a fundamental mechanism to generate complexity in multicellular organisms. In the retina — and the rest of the central nervous system — the major classes of

neurons (e.g. inhibitory *vs.* excitatory; projection *vs.* local) can commonly be divided into specialized subtypes with unique roles and functions. The creation and maintenance of differences between cell subtypes relies predominantly on transcriptional regulation (*Arendt et al., 2016*). Vertebrate photo-receptors — the primary sensors of the visual system — can be classified into multiple subtypes, with a first division between rods and cones, where the high sensitivity of rods is crucial to support vision in dim light and the high adaptability of cones allows signaling throughout the day (*Donner, 1992*). Cones can be further subdivided into several subtypes which differ in spectral sensitivity, morphology, density across the retina, and wiring. Our aim is to identify factors involved in controlling the diversity of photoreceptors, by exploiting technical advantages of the zebrafish model.

The zebrafish retina contains a diverse set of photoreceptor subtypes that is evolutionarily ancient (*Baden and Osorio, 2019*) and includes rods and four cone subtypes. These subtypes can be readily differentiated by their morphology and spectral sensitivity: rods express rhodopsin (Rho, $\lambda_{max}$ = 501 nm), UV cones correspond to the short single cones that express an opsin with peak sensitivity at ultraviolet wavelengths (Opn1sw1, $\lambda_{max}$ = 355 nm); S cones correspond to the long single cones that express an opsin with peak sensitivity at short wavelengths (Opn1sw2, $\lambda_{max}$ = 415 nm); M and L cones are arranged as pairs, where the accessory member corresponds to the M cone, which expresses opsins with peak sensitivities in mid wavelengths (Opn1mw1 - 4, $\lambda_{max}$ = 467–505 nm) and the principal member corresponds to the L cone which expresses opsins with peak sensitivities at long wavelengths (Opn1lw1 - 2, $\lambda_{max}$ = 548–558 nm) (*Chinen et al., 2003*; *Endeman et al., 2013*). Most mammals lost some of this diversity, and only preserved rods and cone subtypes related to the UV and L cones (*Baden and Osorio, 2019*; *Musser and Arendt, 2017*). In addition to differences in morphology and opsin expression, photoreceptor subtypes have distinct wiring with retinal circuits (*Li et al., 2009*; *Li et al., 2012*), differences in density (*Allison et al., 2010*), mitochondrial morphology (*Giarmarco et al., 2020*), and are specialized for particular aspects of vision (*Orger and Baier, 2005*; *Yoshimatsu et al., 2020*; *Yoshimatsu et al., 2021*). Such differences mainly arise from differences in gene expression, ultimately controlled through transcriptional regulation (*Arendt et al., 2016*). Our study seeks to provide the resources and methods required to efficiently identify genes involved in supporting specializations between photoreceptor subtypes, with a focus on transcription factors. The study is divided into four sections. First, we obtain deep and high-quality transcriptomic profiles (RNA-seq) of the five zebrafish photoreceptor subtypes. Second, we explore this RNA-seq dataset and identify multiple transcription factors that could potentially regulate subtype-specific photoreceptor functions. Third, we show that a CRISPR-based F0-screening approach (*Hoshijima et al., 2019*; *Kroll et al., 2021*) is a reliable platform to test the function of these transcription factors. We benchmark our screening method by replicating known phenotypes of *foxq2* and *nr2e3* mutants: F0 larvae that carry mutations in *foxq2* lose S cones, while those that carry mutations in *nr2e3* lose rods (*Ogawa et al., 2021b*; *Xie et al., 2019*). Subsequently, we explore the role of four additional transcription factors with no known function (Skor1a, Sall1a, Lrrfip1a, Xbp1) and find that they are not critical to generate photoreceptor subtypes. Finally, we demonstrate the potential of this platform by describing novel roles for the transcription factors Tbx2a and Tbx2b.We find that the generation of UV cones requires both Tbx2a and Tbx2b, and that Tbx2a and Tbx2b, respectively, maintain the identity of L cones and S cones by repressing M-opsin expression.

## Results

### Transcriptomic analysis of adult zebrafish photoreceptors

Identifying transcription factors that regulate subtype-specific photoreceptor function is critical for understanding how differences between cell subtypes is controlled. RNA-seq is a powerful way to identify novel genes expressed in cell subtypes. Although RNA-seq approaches have been used to identify genes differentially expressed between photoreceptor subtypes in many species, the limited transcriptome depth derived from single-cell techniques (*Macosko et al., 2015*) constitutes a barrier in the reliable detection of transcription factors, which are frequently expressed at low levels (*Wang et al., 2021*). To obtain a deep, high-quality RNA-seq dataset from zebrafish photoreceptors, we used well-characterized transgenic lines that express fluorescent proteins in each subtype with high speci-ficity, including rods — *Tg(xOPS:GFP)*, UV cones — *Tg(opn1sw1:GFP)*, S cones — *Tg(opn1sw2:GFP)*, M cones — *Tg(opn1mw2:GFP)* and L cones — *Tg(thrb:tdTomato)* (*Figure 1A* and *Table 1*; *Fadool,*

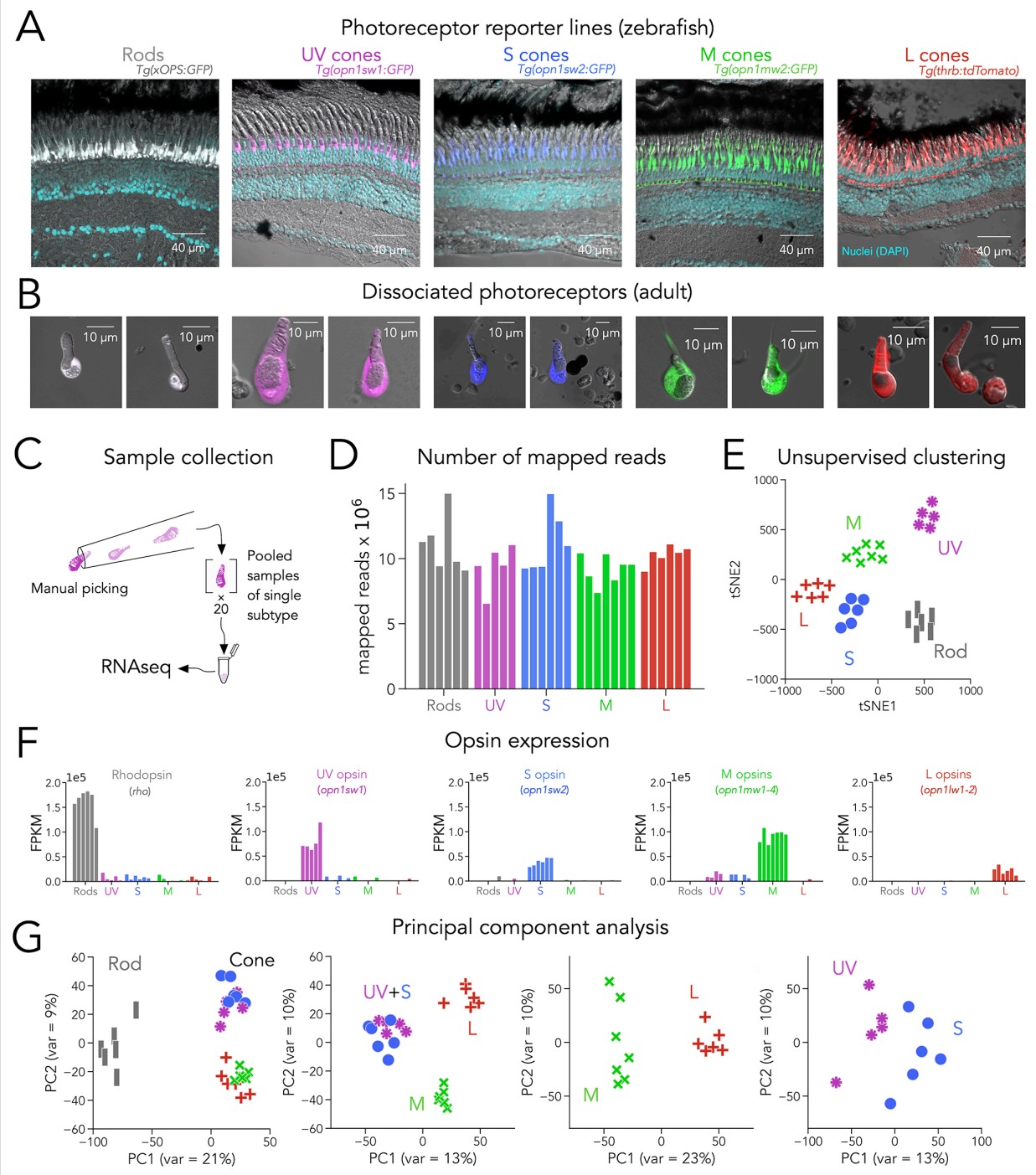

**Figure 1.** Transcriptomic profiling (RNA-seq) of zebrafish photoreceptors. (**A**) Confocal images of fixed adult zebrafish retinal cross-sections, from transgenic reporter lines used to identify photoreceptor subtypes. Reporter expression is exclusive to the outer retina, and each line labels a single photoreceptor subtype with unique morphology, including rods (grey), UV cones (magenta), S cones (blue), M cones (green), and L cones (red). The inner retinal layers can be distinguished in the overlayed nuclear stain (DAPI, cyan) and transmitted DIC image (grey). (**B**) Confocal images of dissociated and live photoreceptors of each subtype, identified by fluorescent reporter expression. Photoreceptors have preserved outer segments and identifiable mitochondrial bundles. (**C**) Sample collection method. After dissociation, 20 healthy photoreceptors of a single subtype were identified by fluorescence and manually picked with a glass micropipette and pooled as a single RNA-seq sample. (**D**) High transcriptome depth shown by the number of reads successfully mapped to the zebrafish genome (GRCz11); bars represent individual RNA-seq samples. (**E**) Clustering using t-distributed stochastic neighbor embedding (tSNE) correctly separates samples by their original subtype; symbols represent individual RNA-seq samples. (**F**) Plots of opsin expression show high counts for the appropriate opsin in each sample (in fragments per kilobase per million reds or FPKM) and low-to-negligible

*Figure 1 continued on next page*

counts of other opsins. For M-opsin quantification, we calculated the sum of counts for *opn1mw1*, *opn1mw2*, *opn1mw3* and *opn1mw4* and for L-opsin quantification, we summed counts for *opn1lw1* and *opn1lw2*. (**G**) Iterative principal component analysis (PCA) shows that differences in gene expression separate rods and cones (first panel), and UV/S cones from M/L cones (second panel). M and L cones can also be distinguished by a single principal component (third panel), while separation of UV and S cones is more difficult (fourth panel).

**Figure supplement 1.** Expression of phototransduction genes.

**Figure supplement 2.** Differentially expressed genes between photoreceptor subtypes.

**Figure supplement 3.** Comparison of RNA-seq datasets across studies.

*2003*; *Suzuki et al., 2013*; *Takechi et al., 2003*; *Tsujimura et al., 2007*), to manually collect pools of photoreceptors of a single subtype under epifluorescence (*Kunze, 2017*). Manual collection allowed us to focus on fluorescent and healthy photoreceptors, with intact outer segments, cell bodies, and mitochondrial bundles, and to avoid cellular debris and other contaminants (*Figure 1B*).

For each sample, we collected pools of 20 photoreceptors of a single subtype derived from a single adult retina. After collection, we isolated mRNA and generated cDNA libraries for sequencing using SMART-seq2 technology (*Figure 1C*). In total, we acquired 6 rod samples and 5 UV-cone, 6 S-cone, 7 M-cone and 6 L-cone samples. On average, we were able to map 86.4% of reads to the zebrafish genome (*GRCz11*; range: 76.3–90.4%), corresponding to 10.19 million±1.77 million mapped reads per sample (mean ± s.d.) and to an average of 7936 unique genes per sample (range: 5508–10,420) (*Figure 1D*). This high quantity of reads and unique genes demonstrates that our technique provides substantially deep transcriptomes — especially when compared to single-cell droplet-based techniques where the number of reads per cell is on average below 20,000, corresponding to more than 2000-fold differences in depth (*Hoang et al., 2020*; *Macosko et al., 2015*; *Ogawa and Corbo, 2021a*). Using unsupervised clustering (t-distributed Stochastic Neighbor Embedding or tSNE), we found that samples correctly clustered by the subtype they were derived from. Proper clustering provides evidence that differences in gene expression captured in our RNA-seq data stem mainly from distinctions between photoreceptor subtypes (*Figure 1E*).

The expression of opsin genes is unique between photoreceptor subtypes and, under normal conditions, a reliable marker of each subtype. Consistent with this idea, each sample had a high number of reads for the appropriate opsin. Namely, rods had high reads for *rho* (or Rhodopsin), UV cones for *opn1sw1* (or UV opsin), S cones for *opn1sw2* (or S opsin), and L cones for *opn1lw1* and *opn1lw2* (which encode two different L opsins). Four different genes — *opn1mw1* to *opn1mw4* — encode M opsins, and their expression in M cones is influenced by retinal region (*Tsujimura et al., 2007*). Because we used *Tg(opn1mw2:GFP)* as an M-cone reporter line, we detected the highest reads for *opn1mw2* as well as *opn1mw3*, both normally expressed in central-to-dorsal retina (*Figure 1—figure supplement 1*). In contrast, we did not detect significant expression of *opn1mw1* or *opn1mw4* in our M-cone samples (see Discussion). In addition, samples only had high expression of the opsin from the reporter line the sample was derived from and low expression of other opsins, corroborating the purity of our samples (*Figure 1F*). We also found that reads for phototransduction genes were high and consistent with the known differences in gene expression between rods and cones (e.g. *gnat1*, rod transducin,

**Table 1.** Zebrafish transgenic lines.

| Label | Transgenic line | Reference |
|---|---|---|
| Rods | *Tg(xOPS:GFP)[fl1Tg]* | *Fadool, 2003* |
| UV cones | *Tg(–5.5opn1sw1:GFP)[kj9Tg]* | *Takechi et al., 2003* |
| UV cones | *Tg(opn1sw1:nfsB-mCherry)[q28Tg]* | *Yoshimatsu et al., 2014* |
| S cones | *Tg(–3.5opn1sw2:GFP)[kj11Tg]* | *Takechi et al., 2008* |
| S cones | *Tg(opn1sw2:nfsB-mCherry)[q30Tg]* | *Yoshimatsu et al., 2014* |
| M cones | *Tg(opn1mw2:GFP)[kj4Tg]* | *Tsujimura et al., 2007* |
| L cones | *Tg(thrb:tdTomato)[q22Tg]* | *Suzuki et al., 2013* |

had high reads only in rod samples while *gnat2*, cone transducin, had high reads in all cone samples) and between cone subtypes (e.g. expression of *arr3a* in M and L cones and *arr3b* in UV and S cones *Figure 1—figure supplement 1*; *Ogawa and Corbo, 2021a*; *Renninger et al., 2011*). To expand our analysis to other genes, we first used principal component analysis (PCA) as an unbiased approach to determine how variability in gene expression defines photoreceptor subtypes. PCA revealed that most of the differences in gene expression were between rods and cones (*Figure 1G*, left panel). When cones were considered separately, the biggest differences in gene expression arose from two groupings: UV and S cones vs. M and L cones. Subsequent analysis revealed a clear separation of M and L cones, with UV and S cones showing the least differences (*Figure 1G*). Guided by this analysis, we performed differential gene-expression analysis by making pairwise comparisons following the directions of the principal components, revealing a diverse set of ~3000 differentially expressed genes, many of unknown function in photoreceptors (*Figure 1—figure supplement 2* and *Supplementary file 1*).

In summary, our manual, cell-type specific, SMART-seq2-based approach yielded high-quality zebrafish photoreceptor transcriptomes, with low contamination and ~2000-fold more depth than published single-cell RNA-seq studies in the retina, and thus has a particularly high signal-to-noise ratio for differential gene-expression analysis (*Hoang et al., 2020*; *Macosko et al., 2015*; *Ogawa and Corbo, 2021a*; *Peng et al., 2019*; *Figure 1—figure supplement 3*). As exemplified by phototransduction proteins (and transcription factors below), our dataset is in good agreement with current knowledge of photoreceptor-expressed genes. In addition, it uncovered novel and unexplored differences in gene expression between photoreceptor subtypes. This RNA-seq dataset constitutes a useful resource to explore genes that are generally or differentially expressed by photoreceptor subtypes which could be involved in multiple aspects of photoreceptor biology, especially when integrated with other relevant studies (see discussion). For this reason, we have made our dataset openly available and easy to explore through an online interactive plotter (https://github.com/angueyraLab/drRNAseq/), and integrated it with other available zebrafish photoreceptor datasets (*Hoang et al., 2020*; *Ogawa and Corbo, 2021a*; *Sun et al., 2018*). Our subsequent analyses center on transcription factors.

## Expression of transcription factors in zebrafish photoreceptors

One of our main interests is to understand the transcriptional regulation that leads to diverse photoreceptor subtypes. Therefore, we isolated all RNA-seq reads that could be mapped to transcription factors and detected significant expression of 803 transcription factors. When ranked by average expression levels across all samples, *neurod1* was revealed as the transcription factor with the highest expression by ~fivefold (*Figure 2A*). High expression in adult photoreceptors suggests that Neurod1 plays a role in the mature retina in addition to its well-established roles in development and regeneration (*Ochocinska and Hitchcock, 2009*; *Taylor et al., 2019*; *Thomas et al., 2012*). Among the 100 most highly expressed transcription factors, we identified genes well known to be critical during photoreceptor development, including *crx, otx5, rx1, rx2, nr2e3, six6b, six7, meis1b, egr1, foxq2,* and *thrb* (*Figure 2A*, blue bars; *Furukawa et al., 1997*; *Shen and Raymond, 2004*; *Ji et al., 2018*; *Swaroop et al., 2010*; *Ogawa et al., 2015*; *Ogawa et al., 2019*; *Erickson et al., 2010*; *Heine et al., 2008*; *Ogawa et al., 2021b*; *Zhang et al., 2013*). Of the remaining transcription factors in this short list, only a limited number of studies have explored their function, despite their high expression on our dataset (*Figure 2A*, grey bars; *Bhootada et al., 2016*; *Fotaki et al., 2013*; *Lenkowski et al., 2013*; *McLaughlin et al., 2018*; *Taylor et al., 2019*; *Giarmarco et al., 2020*; *Gross et al., 2005*; *Mollema et al., 2011*; *Cheng et al., 2004*). Furthermore, many of these genes have not been previously studied or identified as expressed in photoreceptors (*Figure 2A*, black bars), suggesting that our current knowledge on the control of genetic programs in photoreceptors remains incomplete.

Next, we used PCA to understand how the expression of these 803 transcription factors differs between photoreceptor subtypes. Like our whole-transcriptome analysis, we found that most of the differences in transcription-factor expression can be attributed to differences between rods and cones (*Figure 2B*). By performing pairwise comparisons of transcription factors based on rod *versus* cone expression, we identified three relevant groups: (1) consistent expression across all subtypes, (2) rod-enriched and (3) cone-enriched (*Figure 2C*). Consistent with previous studies, expression of *crx* and *otx5* was similar across subtypes (*Furukawa et al., 1997*; *Shen and Raymond, 2004*); *nr2e3, samd7* and *samd11* showed clear rod-enrichment (*Kubo et al., 2021*; *Omori et al., 2017*; *Oh et al., 2008*)

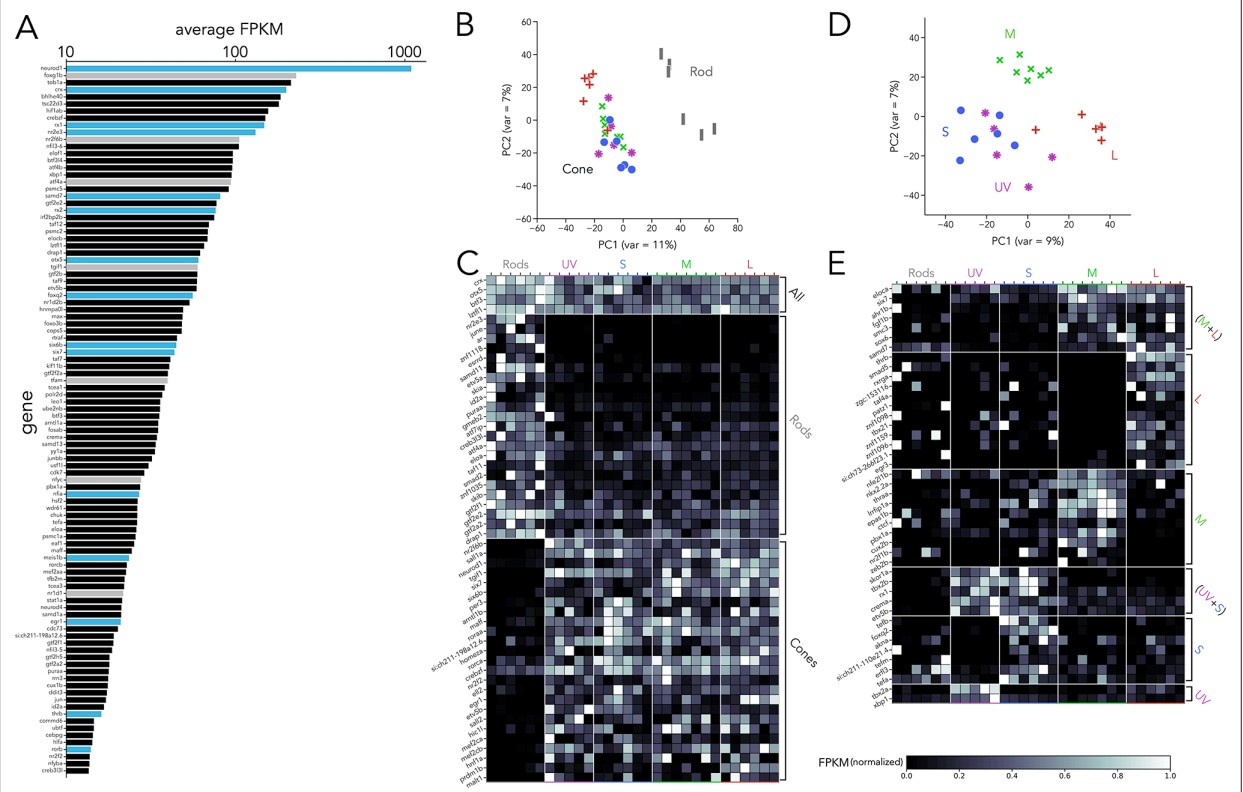

**Figure 2.** Transcription factor expression in zebrafish adult photoreceptors. (**A**) Top 100 transcription factors ranked by average expression across all samples, displayed on a logarithmic scale. Genes highlighted in blue are known to be critical in photoreceptor progenitors or during photoreceptor development. Some limited information about function in photoreceptors exists for genes highlighted in grey. (**B**) Principal component analysis (PCA) of transcription factor expression shows that differences between rods and cones are the main source of variance. (**C**) Heatmap showing differential expression of transcription factors between rods and cones and divided into three groups: consistently expressed by all photoreceptors, enriched in rod samples, and enriched in cones samples. Grey values indicate expression level normalized by the maximal value in each row. Rod- and cone-enriched genes have been arranged by degree of enrichment. (**D**) PCA of cone samples shows that the largest differences in expression separate L cones (PC1) and M cones (PC2), while separation of UV and S cones is more difficult. (**E**) Heatmap of transcription factors differentially expressed in cone subtypes, divided into six relevant groups. Full list of differentially expressed transcription factors available in *Supplementary file 2*, *Supplementary file 3*.

The online version of this article includes the following figure supplement(s) for figure 2:

**Figure supplement 1.** Transcripts from candidate transcription factors are detected in photoreceptors.

**Figure supplement 2.** Expression of candidate transcription factors is active during photoreceptor development.

while *six6a*, *six6b*, *six7*, *sall1a*, and *neurod1* showed cone-enrichment (***Ogawa et al., 2019***; ***Ochocinska and Hitchcock, 2009***; ***Ogawa et al., 2015***; ***Lonfat et al., 2021***). By expanding our analysis beyond previously characterized genes, our dataset revealed a total of 75 transcription factors with significant differential expression between rods and cones, many of which have no described function in photoreceptors (***Figure 2C*** and Supplementary Data 2).

We next examined the variance in transcription-factor expression between cone subtypes. PCA revealed that both L and M cones could be distinguished by differences in transcription-factor expression alone, while UV and S cones again showed the fewest differences (***Figure 2D***). By analyzing cone subtypes, we found a total of 47 differentially expressed transcription factors. Seven transcription factors were significantly enriched in both L and M cones compared to UV and S cones (***Figure 2E***) and included *ahr1b* — a gene associated with Retinitis Pigmentosa in humans (***Zhou et al., 2018***) — and *six7* — known to be involved in cone progenitor development and survival (***Ogawa et al., 2015***). Twelve were enriched in L cones, including *thrb* — known to be critical to generate L-cones across vertebrates (***Ng et al., 2001***; ***Suzuki et al., 2013***) — and *rxrga* — a regulator of L-opsin expression in mouse (***Roberts et al., 2005***). Amongst the ten M-cone enriched transcription factors, we identified *thraa* — another thyroid hormone receptor, confirmed to be expressed by photoreceptors (***Volkov***

**Table 2.** crRNA sequences.

| Gene | gRNA | Efficiency |
|------|------|------------|
| *foxq2* | TCATTTCTGGGCAATTCACCCGG<br>CCCATCCGTTATTGTGCTTCCCG | 95% (21/22) |
| *nr2e3* | CCTGGAAAGGTCCTGAACACGGG<br>TATGGAATATACGCTTGCAACGG | 100% (16/16) |
| *tbx2a* | TAACGATATGAAACCTGGGTTGG<br>GACAGCTATAAAATCGGTCTCGG<br>GGCTCTAACGATATGAAACCTGG | 91% (30/33) |
| *tbx2b* | TATCGTTGGCTCTCACAATATGG<br>CAAGGTATGTACCCATATTTTGG<br>CGGAAGCTTCAGAATATCGTTGG | 83% (19/23) |
| *skor1a* | CCTCTGCAAATCCTTTCTCGGGG<br>CGCCAGGTACAATAGCTCCAGGG<br>GTTTCACACGAGTGCGCCTGGGG | 94% (17/18) |
| *xbp1* | AAATGGTCGTAGTTACAGCAGGG<br>GCTTCGACCGGCGCGACACAGGG<br>CCGGCGCGACACAGGGCGGGTAC | 93% (13/14) |
| *sall1a* | CCCACTCAGTGGTGTTGGAACTG<br>GGAACTGGCCATGGAACGCTGGG<br>ATGGAACGCTGGGAAGCACTGGG | 96% (23/24) |
| *lrrfip1a* | CCGTTTGGCAGCGAAGAGAGCGG<br>CCCTGCAGGCTGAAGCCCGTTTG<br>TGAGATCAGAATGAAAGAACTGG | 100% (16/16) |

*et al., 2020*) — and *lrrfip1a*. A small group of just five genes was enriched in both UV and S cones compared to L and M cones and included *tbx2b* and *skor1a*. Seven transcription factors were enriched in S cones — including *foxq2* — and two were enriched in UV cones — *tbx2a* and *xbp1* (**Figure 2E** and Supplementary Data 3). To validate our RNA-seq, we used a fluorescent in situ hybridization assay and detected expression in photoreceptors of several transcription factors identified through this analysis (**Figure 2—figure supplement 1**). Reanalysis of existing RNA-seq datasets confirms that the expression of the identified transcription factors is active during retinal development (**Figure 2—figure supplement 2**).

In summary, our transcriptomic analysis is in good agreement with our current knowledge of transcription factor expression in photoreceptors. Additionally, it reveals novel patterns of expression between photoreceptor subtypes. Notably, a considerable fraction of these transcription factors has no clear function in photoreceptors in zebrafish or in other species, making them clear targets for follow-up studies aimed at understanding how differences between photoreceptor subtypes are transcriptionally regulated.

## F0 screening as a reliable platform to explore transcriptional control of subtype-specific functions in photoreceptors

Our dataset revealed an extensive collection of transcription factors with differential expression between photoreceptor subtypes, which are likely to control subtype-specific functions. Akin to *terminal selectors*, some of these transcription factors could be potentially involved in the generation and maintenance of photoreceptor subtypes (**Arendt et al., 2016**; **Hobert, 2008**). Given the high number of candidates and our limited knowledge on their function, we sought to establish methods to efficiently produce loss-of-function mutations and evaluate subtype-specific phenotypes, through the use of CRISPR-based F0 screening (**Hoshijima et al., 2019**; **Kroll et al., 2021**). We injected single-cell zebrafish embryos with Cas9 protein and 2 or 3 guide RNAs (gRNAs) targeting a gene of interest. All guides were tested to ensure a high rate of mutations (**Table 2**). At 5 days post-fertilization (5 dpf) we assessed phenotypes in injected F0 larvae, which are genetic mosaics (some cells may not carry mutations and mutations are not identical in every cell). In our analyses, we screened F0 larvae for defects in the generation and maintenance of photoreceptor subtypes, by quantifying photoreceptor densities in the central retina using subtype-specific reporter lines (**Table 1**). All analyses correspond

to F0 larvae that have been genotyped to confirm mutations in the targeted gene (see Materials and methods).

To benchmark this F0 screen in the context of the generation of photoreceptor subtypes, we first targeted two genes with subtype-specific expression in our RNA-seq that are known to be involved in this process — Foxq2 and Nr2e3. Among transcription factors, *foxq2* is expressed at relatively high levels, ranking 33<sup>rd</sup> amongst the top enriched transcription factors (*Figure 2A*). It is specifically enriched in S cones, with negligible expression in other photoreceptor subtypes (*Figure 3A*). Loss-of-function of *foxq2* mutants are characterized by a complete loss of S cones and S-opsin expression, and a slight increase in M-opsin expression (*Ogawa et al., 2021b*). For our F0 screen, we designed two gRNAs targeted against the DNA-binding forkhead domain of Foxq2 (*Yu et al., 2003*). Compared to wild-type controls, *foxq2* F0 mutant larvae displayed a marked decrease of ~85% in the density of S cones (*Figure 3B* and *Figure 3—figure supplement 1*). Consistent with the slight increase in M-opsin expression reported in germline loss-of-function mutants, we also found a small but significant increase of ~24% in the density of M cones in foxq2 F0 mutants. In contrast, the densities of rods, UV cones, and L cones in *foxq2* F0 mutants did not show any significant differences (*Figure 3C*). Quantification of overall cone density — using nuclear staining — did not reveal significant differences in *foxq2* F0 mutants compared to control (*Figure 3—figure supplement 2*).

As a second positive control, we created mutations in *nr2e3* — a rod enriched-gene (*Figure 3D*), known to be critical for the generation of rods in vertebrates (*Cheng et al., 2006*; *Forrest and Swaroop, 2012*; *Oh et al., 2008*) — by injecting two gRNAs targeted against exon 1. As observed in germline *nr2e3* mutants (*Xie et al., 2019*), *nr2e3* F0 mutants have a pronounced loss of ~80% of rods (*Figure 3E*). Interestingly, in *nr2e3* F0 mutants we also identified a ~25% decrease in UV-cone densities — which has not been previously reported — suggesting an unrecognized role of Nr2e3 in cone development (see discussion) (*Figure 3F and G*). The loss of UV cones is reflected in a decrease in overall cone density in *nr2e3* F0 mutants (*Figure 3—figure supplement 2*).

The close agreement between germline mutants and *foxq2* and *nr2e3* F0 mutants demonstrates that our approach is reliable (phenotypes are clear and quantifiable), flexible (mutations were created using any relevant combination of transgenic lines) and efficient in terms of cost and labor (a gene can be evaluated in less than a month by a single person, without significantly increasing burden in animal care). This motivated us to screen new and poorly characterized candidate genes with differential expression across photoreceptors, including *skor1a*, *sall1a*, *lrrfip1a,* and *xbp1*.

Across multiple photoreceptor transcriptomic datasets, including ours, the expression of *skor1a* is restricted to UV and S cones (*Figure 3—figure supplement 3A*, *Figure 1—figure supplement 3E*), and high in early stages of cone development (*Figure 2—figure supplement 2G*; *Hoang et al., 2020*; *Ogawa and Corbo, 2021a*). In humans, MEIS1 regulates expression of *SKOR1* (*Catoire et al., 2018*). MEIS1 is key for the proper regulation of retinal progenitors across vertebrates (*Erickson et al., 2010*; *Heine et al., 2008*), making Skor1a a candidate factor that could be involved in the specification of UV and S cones (*Ogawa and Corbo, 2021a*).In disagreement with this hypothesis, we find that *skor1a* F0 mutants have normal UV and S cone densities and normal total cone densities (*Figure 3—figure supplement 3B–D*). The cone-specific gene *sall1a* is hypothesized to be involved in rod *vs.* cone differentiation in chicken (*Enright et al., 2015*; *Ghinia-Tegla et al., 2021*; *Lonfat et al., 2021*). Yet, we found that *sall1a* F0 mutants have normal rod densities and no disturbance of the cone mosaic or total cone density (*Figure 3—figure supplement 4*). Similarly, F0 mutants of the M-cone enriched gene *lrrfip1a*, have normal M-cone densities (*Figure 3—figure supplement 5*), and F0 mutants of the UV-cone enriched gene *xbp1*, have no appreciable changes in UV-cone or other photoreceptor-subtype densities (*Figure 3—figure supplement 6*). These results suggest that these four transcription factors are not critical for the generation of photoreceptor subtypes. They are likely to play other subtype-specific roles that warrant future investigations.

## Tbx2 plays multiple roles in the generation and maintenance of photoreceptor subtypes

### Tbx2a and Tbx2b are independently required for the generation of UV cones

To further expand our F0 analysis, we explored the role of Tbx2 in the generation of photoreceptor subtypes. Tbx2 is known to be differentially expressed in cones of many species, including cichlids

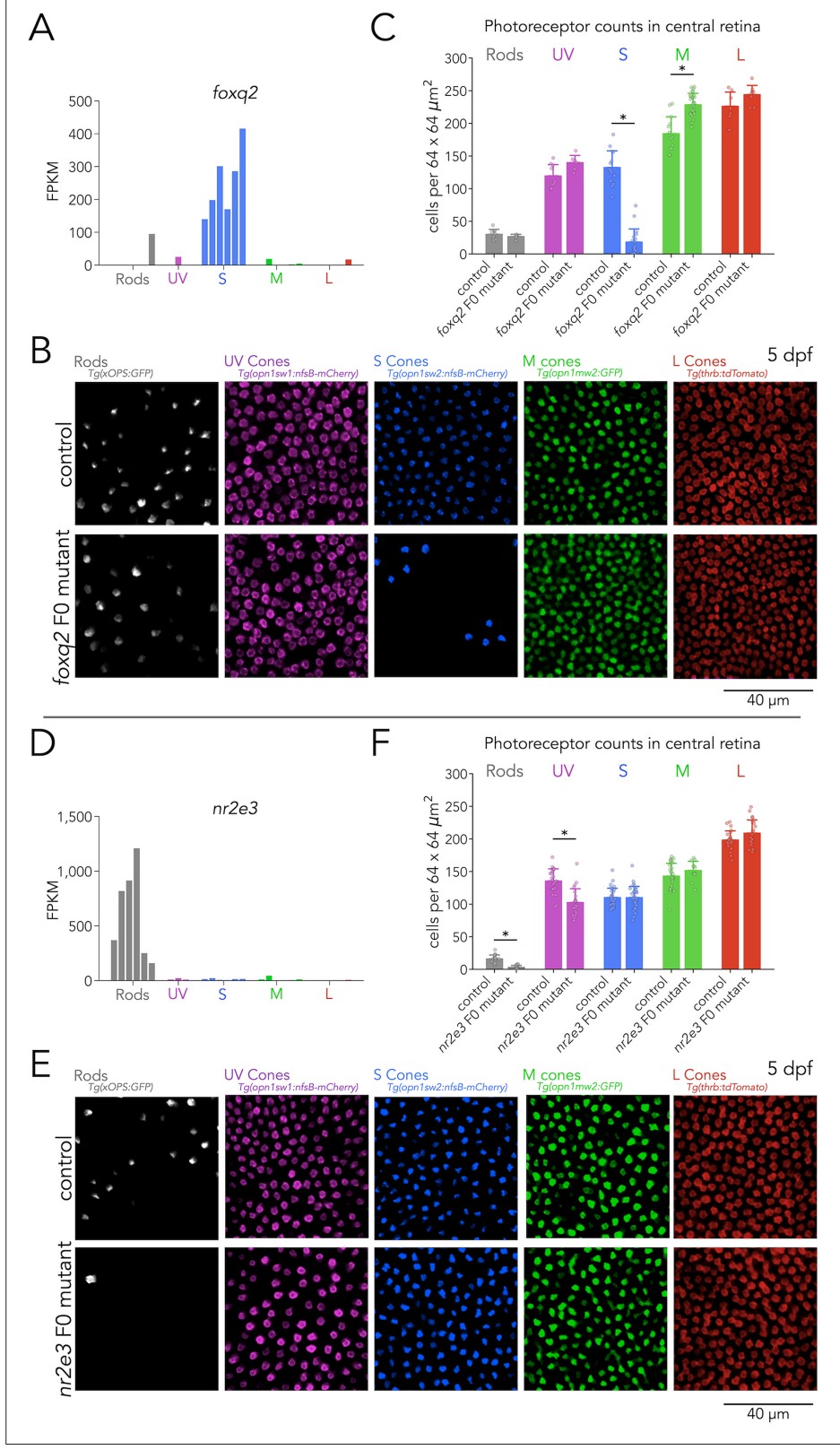

**Figure 3.** Foxq2 is required for generation of S cones and Nr2e3 for generation of rods. (**A**) Expression of *foxq2* shows clear S-cone specificity. (**B**) Mutations in *foxq2* cause a loss of S cones. Representative confocal images of the central retina of control (top row) and *foxq2* F0 mutant (bottom row) larvae at 5 dpf. Each column corresponds to a transgenic line that labels a unique photoreceptor subtype, pseudo-colored according to photoreceptor

*Figure 3 continued on next page*

*Figure 3 continued*

subtype. (**C**) Quantification of photoreceptors in control and *foxq2* F0 mutant larvae. Bars represent averages, error bars correspond to standard deviations, and markers correspond to individual retinas. There is a significant ~86% reduction in S cones in *foxq2* F0 mutants compared to *wildtype (wt)* controls (Kruskal-Wallis H=22.93, p=1.6.×$10^{-6}$,, $n_{wt}$ = 14, $n_{foxq2}$=18) and a smaller but significant ~24% increase in the density of M cones (Kruskal-Wallis H=17.55, p=2.8.×$10^{-5}$, $n_{wt}$ = 13, $n_{foxq2}$=28). We found no significant differences in the densities of rods (Kruskal-Wallis H=0.784, p=0.376, $n_{wt}$ = 9, $n_{foxq2}$=6), UV cones (Kruskal-Wallis H=3.562, p=0.059, $n_{wt}$ = 9, $n_{foxq2}$=6), or L cones (Kruskal-Wallis H=2.267, p=0.132, $n_{wt}$ = 7, $n_{foxq2}$=8). (**D**) Expression of *nr2e3* shows enrichment in rods. (**E**) Mutations in *nr2e3* cause a loss of rods. Representative confocal images of the central retina of control (top row) and *nr2e3* F0 mutant (bottom row) larvae at 5 dpf. (**F**) Quantification of photoreceptors in control and *nr2e3* F0 mutant larvae. Bars represent averages, error bars correspond to standard deviations, and markers correspond to individual retinas. There is a significant ~80% reduction in rods in *nr2e3* F0 mutants compared to controls (Kruskal-Wallis H=26.987, p=2.0 × $10^{-7}$, $n_{wt}$ = 19, $n_{nr2e3}$=19), a smaller but significant ~25% reduction in UV cones (Kruskal-Wallis H=18.77, p=1.5 × $10^{-5}$, $n_{wt}$ = 24, $n_{nr2e3}$=24). We found no significant differences in the densities of S cones (Kruskal-Wallis H=0.024, p=0.87, $n_{wt}$ = 30, $n_{nr2e3}$=32), M cones (Kruskal-Wallis H=1.61, p=0.205, $n_{wt}$ = 30, $n_{nr2e3}$=12), or L cones (Kruskal-Wallis H=2.407, p=0.12, $n_{wt}$ = 24, $n_{nr2e3}$=22).

The online version of this article includes the following figure supplement(s) for figure 3:

**Figure supplement 1.** Mutations in *foxq2* cause a decrease in S opsin positive photoreceptors.

**Figure supplement 2.** Assessment of cone density in *foxq2* and *nr2e3* F0 mutants.

**Figure supplement 3.** Skor1a is not required for UV-cone or S-cone specification.

**Figure supplement 4.** Sall1a is not required for photoreceptor specification.

**Figure supplement 5.** Lrrfip1a is not required for M-cone specification.

**Figure supplement 6.** Xbp1 is not required for photoreceptor specification.

(*Sandkam et al., 2020*), chickens (*Yamagata et al., 2021*), squirrels (*Kunze, 2017*), and primates (*Peng et al., 2019*). As a teleost duplicated gene, there are two paralogues of *tbx2* in the zebrafish genome: *tbx2a* and *tbx2b*. Work in zebrafish has shown that Tbx2b is involved in the determination of UV-cone fate (*Alvarez-Delfin et al., 2009*). Our RNA-seq data revealed that both *tbx2a* and *tbx2b* show high expression in UV cones (*Figure 4A*). In addition, we detected significant enrichment of *tbx2a* and *tbx2b* expression in L and S cones, respectively. This expression data suggested that Tbx2 might play unexplored roles in photoreceptors.

We first focused our analysis on the role of Tbx2 in UV cones. For our F0 analysis, we designed 3 gRNAs targeting exon 3 of *tbx2b* or *tbx2a*. In both genes, exon 3 contains critical DNA-binding residues that are completely conserved across vertebrates (*Sinha et al., 2000*). In control larvae at 5 dpf, UV cones are numerous and densely distributed across the retina, while overall rod density is low, with most rods concentrated in the ventral retina and the lowest density in the central retina (*Alvarez-Delfin et al., 2009*; *Yoshimatsu et al., 2014*; *Raymond et al., 2014*). In agreement with previous studies, *tbx2b* F0 mutants had a marked decrease in UV cones (~62%) and an increase in rod density (~3.85-fold) (*Figure 4B and D*; *Alvarez-Delfin et al., 2009*). After replicating the described phenotypes of germline *tbx2b* mutants in *tbx2b* F0 mutants, we examined *tbx2a* F0 mutants. Surprisingly, we found that *tbx2a* F0 mutants displayed the same phenotype as *tbx2b* F0 mutants: a marked loss of UV cones (~59%) and an increase in rods (~1.86-fold) — although the increase in rods was significantly lower in *tbx2a* F0 mutants than in *tbx2b* F0 mutants (*Figure 4C and D*). This difference in rod density is apparent when the number of rods and UV cones are summed. Compared to controls, *tbx2b* F0 mutants have a modest but significant decrease in the sum of rods and UV cones (~14%), and *tbx2a* F0 mutants have a marked loss (~39%) (*Figure 4E*). Additionally, we quantified overall cone density in *tbx2* F0 mutants using nuclear staining with DAPI. As expected, the loss of UV-cone nuclei is readily apparent in both *tbx2a* and *tbx2b* F0 mutants (~66.6% and 69.1%, respectively, corresponding to 129 and 134 fewer UV cones than controls), and both mutants show a significant decrease in total cone density (~9.8% and 12.5% respectively, corresponding to 72 and 91 fewer cones than controls) (*Figure 6—figure supplement 1A–B*). Since this decrease in total cone density is smaller than the loss expected from UV cones, we explored changes in the densities of other cone subtypes (see next section).

To confirm the phenotypes of *tbx2* mutants revealed through imaging of reporter lines, we quantified opsin expression using real-time quantitative PCR (qPCR). We found that, in comparison to

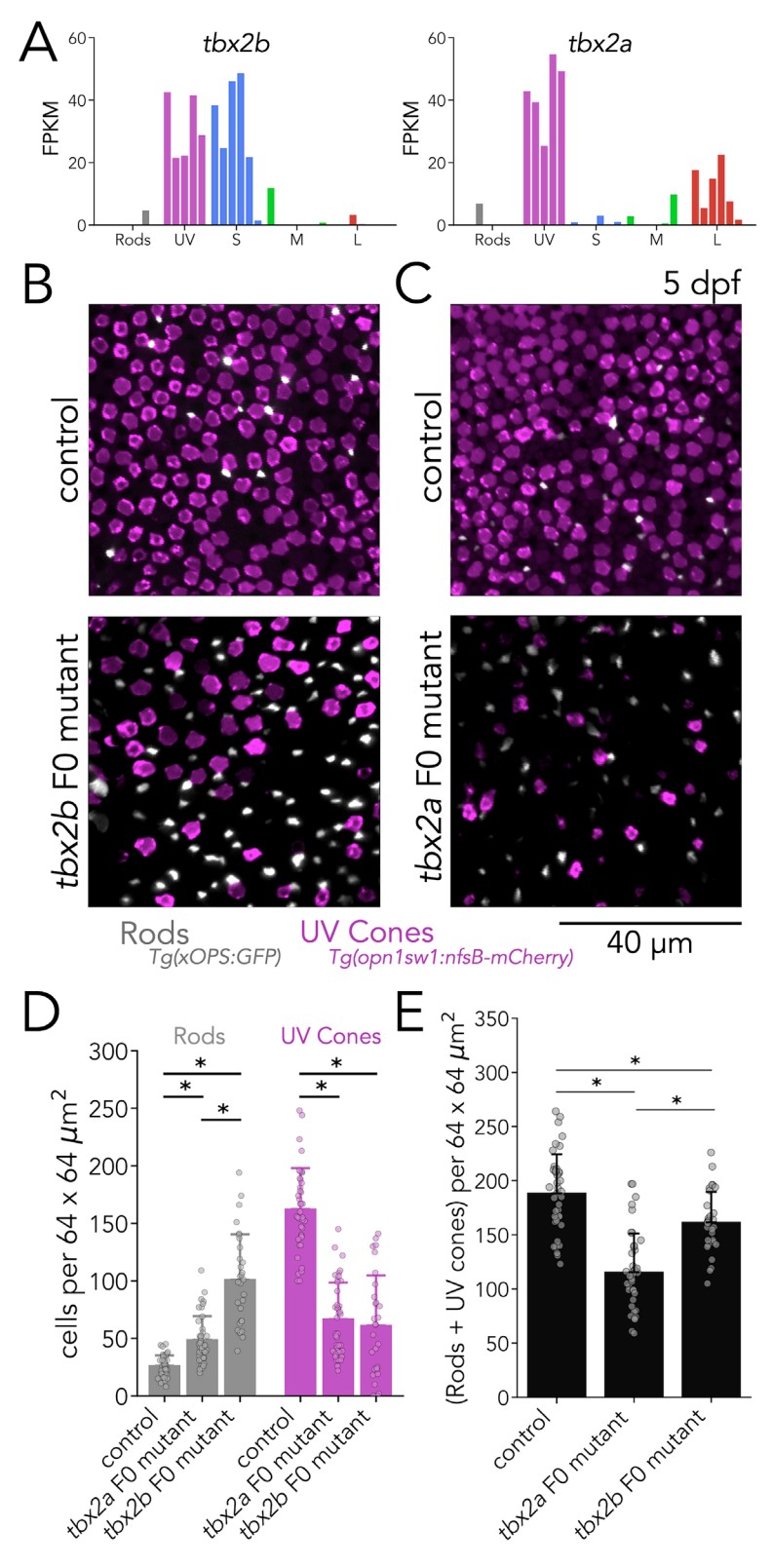

**Figure 4.** Tbx2a and Tbx2b are independently required for generation of UV cones. (**A**) *tbx2b* is expressed by UV and S cones (left), while *tbx2a* is expressed by UV and L cones (right). (**B**) Mutations in *tbx2b* cause a loss of UV cones and an increase in rods. Representative confocal images of the central retina of control and *tbx2b* F0 mutants at 5 dpf, in double transgenic larvae with labeled UV cones (magenta) and rods (grey). (**C**) Mutations in

*Figure 4 continued on next page*

*Figure 4 continued*

*tbx2a* also cause a loss of UV cones and an increase in rods. Representative confocal images of the central retina of control and *tbx2a* F0 mutants at 5 dpf, in the same double transgenic lines. (**D**) Quantification of rods and UV cones in control, *tbx2a* and *tbx2b* F0 mutant larvae. Bars represent averages, error bars correspond to standard deviations, and markers correspond to individual retinas. Compared to controls, both tbx2 F0 mutants have a significant increase in rods (1.86-fold for *tbx2a* and 3.86-fold for *tbx2b*, Kruskal-Wallis H=71.725, p=2.67 × 10$^{-16}$, n$_{wt}$ = 40, n$_{tbx2a}$=40, n$_{tbx2b}$=30; Conover-Iman *posthoc* corrected p-values: control *vs.* *tbx2a* p=2.96 × 10$^{-11}$, control *vs.* *tbx2b* p=4.16 × 10$^{-26}$) but this increase significantly smaller in *tbx2a* F0 mutants (*tbx2a* *vs.* *tbx2b* p=2.25 × 10$^{-10}$); both *tbx2* F0 mutants have a marked decrease in UV cones (58.91% for *tbx2a* and 62.32% for *tbx2b*, Kruskal-Wallis H=66.907, p=2.96 × 10$^{-15}$, n$_{wt}$ = 40, n$_{tbx2a}$=40, n$_{tbx2b}$=29; Conover-Iman *posthoc* corrected p-values: control *vs.* *tbx2a* p=1.71 × 10$^{-19}$, control *vs.* *tbx2b* p=5.41 × 10$^{-19}$); this increase was not significantly different between *tbx2a* and *tbx2b* F0 mutants (*tbx2a* *vs.* *tbx2b* p=1.0). (**E**) Quantification of the sum of rods and UV cones in control, *tbx2a* and *tbx2b* F0 mutants. Bars represent averages, error bars correspond to standard deviations, and markers correspond to individual retinas. Compared to control both *tbx2a* and *tbx2b* mutants have a significant decrease in the sum of rods and UV cones, but this decrease is significantly more pronounced in *tbx2a* F0 mutants (38.81% for *tbx2a* and 14.34% for *tbx2b*, Kruskal-Wallis H=50.156, p=1.29 × 10$^{-11}$, n$_{wt}$ = 40, n$_{tbx2a}$=40, n$_{tbx2b}$=29; Conover-Iman *posthoc* corrected p-values: control *vs.* *tbx2a* p=2.38 × 10$^{-15}$, control *vs.* *tbx2b* p=6.72 × 10$^{-3}$, *tbx2a* *vs.* *tbx2b* p=5.92 × 10$^{-7}$).

The online version of this article includes the following figure supplement(s) for figure 4:

**Figure supplement 1.** Assessment of total cone density in *tbx2a* and *tbx2b* F0 mutants.

**Figure supplement 2.** Mutations in *tbx2a* and *tbx2b* cause multiple changes in opsin expression.

controls, *tbx2b* F0 mutants showed a clear decrease in UV-opsin expression and a significant increase in rhodopsin expression. In comparison, *tbx2a* F0 mutants also showed a clear decrease in UV-opsin expression, but without an increase in rhodopsin expression (***Figure 4—figure supplement 1***). Together our reporter lines and qPCR analyses suggest that, despite 87% protein-sequence similarity and co-expression of the two genes in the same cell, both Tbx2a and Tbx2b are required for the generation of zebrafish UV cones. Loss-of-function of either gene leads to a decrease in UV cones and a concomitant routing of photoreceptor progenitors towards a rod fate (***Alvarez-Delfin et al., 2009***). In *tbx2b* F0 mutants routing towards a rod fate appears to be significantly stronger than in *tbx2a* F0 mutants.

## Tbx2a inhibits M-opsin expression in L cones

After ascertaining the requirement of Tbx2a and Tbx2b in UV-cone generation, we examined whether either of these transcription factors impacted the identity of other photoreceptor subtypes. In addition to expression in UV cones, we detected significant enrichment of *tbx2a* in L cones, albeit with expression levels lower than in UV cones (***Figure 4B***). Furthermore, our qPCR quantification of opsins revealed a significant increase in M-opsin expression in *tbx2a* F0 mutants — specifically of *opn1mws2* (***Figure 4—figure supplement 1***). Based on these results, we tested whether Tbx2a is involved in the control of M-cone or L-cone identity.

To examine M and L cones, we assessed *tbx2a* F0 mutants using an M-cone reporter line, where GFP expression is under direct control of the M-opsin promoter — *Tg(opn1mws2:GFP)* — in combination with an L-cone reporter line — *Tg(thrb:tdTomato)*. In control larvae, the expression of GFP and tdTomato is non-overlapping, reflecting the distinction between M cones and L cones (***Figure 5A***, left). In *tbx2a* F0 mutants, we found a significant but small decrease in the number of L cones (~12%) — identified by their tdTomato expression (***Figure 5B***), and a marked increase in the number of GFP-positive cells (presumptive M cones). Interestingly, in *tbx2a* F0 mutants, many GFP-positive cells co-express tdTomato (L-cone marker) — a phenotype which is not present in control larvae (***Figure 5A***, middle). To quantify this effect, we calculated the fraction of tdTomato-positive L cones with significant GFP expression (see methods). We found that only a small fraction of L cones is double positive in controls (mean ± s.d.: 5.2%±6.0), but a significantly higher fraction is double positive in *tbx2a* F0 mutants (mean ± s.d.: 37.5% ± 18.9%) (***Figure 5C***). This abnormal expression of GFP in L cones in *tbx2a* F0 mutants, combined with the increase in M-opsin expression found in our qPCR analysis, indicates a loss of inhibitory control over the M-opsin promoter. By manually excluding double-positive cells, we counted GFP-only cells to quantify M cone densities and found no significant changes in *tbx2a* F0 mutants (***Figure 6—figure supplement 1A–B***).

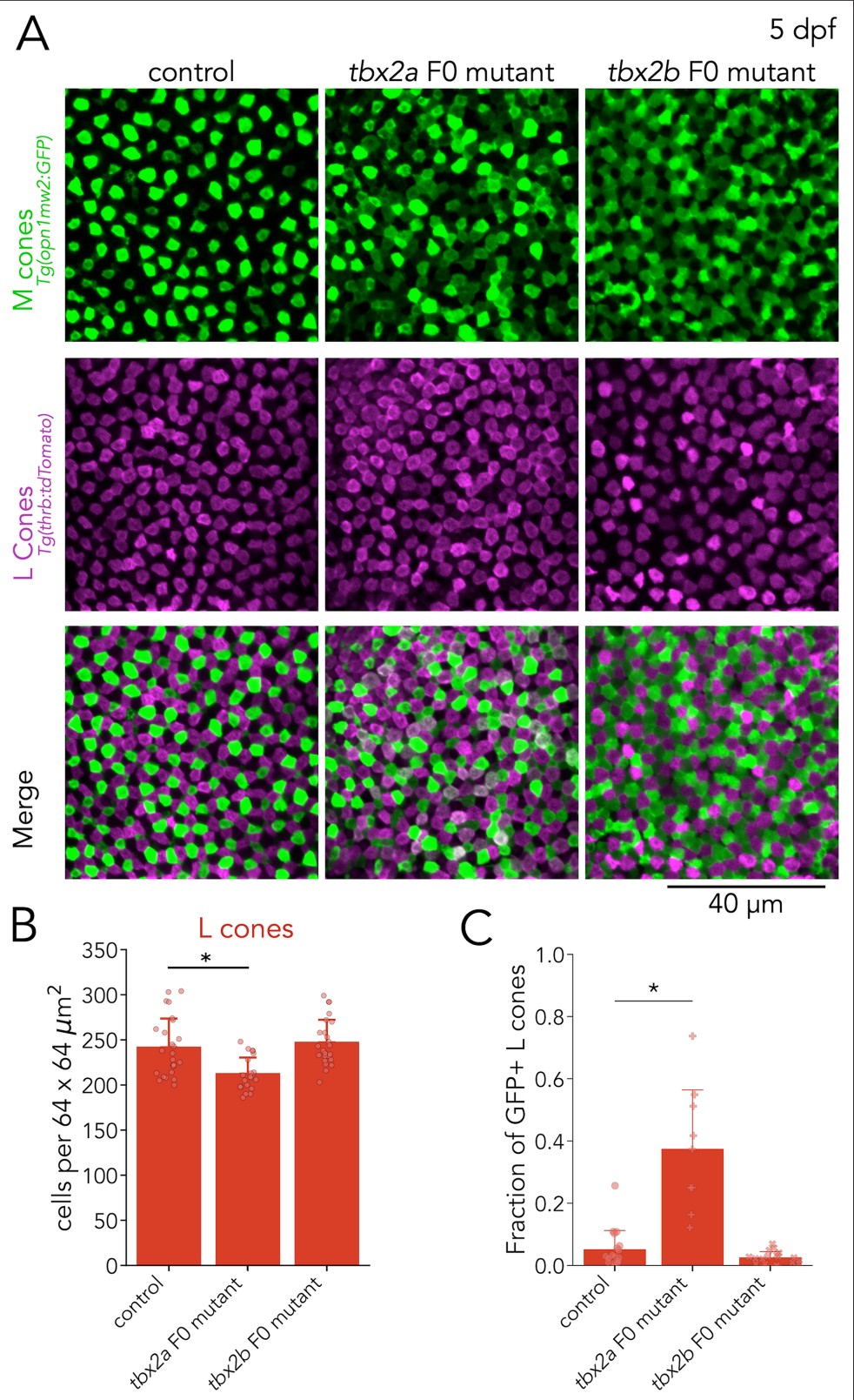

**Figure 5.** Tbx2a inhibits M-opsin expression in L cones. (**A**) Representative confocal images of the central retina of control, *tbx2a* and *tbx2b* F0 mutants at 5 dpf, in double transgenic larvae that label M cones — or M-opsin expressing cells — with GFP (green) and L cones with tdTomato (magenta). Both *tbx2a* and *tbx2b* F0 mutants display an increase in GFP-positive cells. In *tbx2a* F0 mutants, increase in GFP signal is restricted to tdTomato-

*Figure 5 continued on next page*

*Figure 5 continued*

positive cells which appear as double-positive (white) in merged images, while in *tbx2b* F0 mutants, increase in GFP signal is excluded from tdTomato-positive cells, producing a decrease in the space without fluorescence. (**B**) Compared to controls, *tbx2a* F0 mutants have a significant decrease in L cones (12.12% for *tbx2a*, Kruskal-Wallis H=18.264, p=$1.08 \times 10^{-4}$, $n_{wt}$ = 25, $n_{tbx2a}$=21, $n_{tbx2b}$=24; Conover-Iman *posthoc* corrected p-values: control *vs. tbx2a* P=$1.21 \times 10^{-3}$, control *vs. tbx2b* p=0.88, *tbx2a vs. tbx2b* p=$3.98 \times 10^{-5}$) (**C**) Quantification of the fraction of GFP-positive L cones (double positive cells in A) reveals a significant increase only in *tbx2a* F0 mutants (Kruskal-Wallis H=20.821, p=$3.01 \times 10^{-5}$, $n_{wt}$ = 18, $n_{tbx2a}$=9, $n_{tbx2b}$=17; Conover-Iman *posthoc* corrected p-values: control *vs. tbx2a* p=$2.87 \times 10^{-5}$, control *vs. tbx2b* p=0.63, *tbx2a vs. tbx2b* p=$1.13 \times 10^{-6}$).

As a control, we repeated this M-cone and L-cone assessment in *tbx2b* F0 mutants — despite no detectable expression of *tbx2b* in M or L cones (*Figure 3B*). While *tbx2b* F0 mutants also had a significant increase in the number of GFP-positive cells (see next section) (*Figure 5A*, bottom), there were no changes in the number of tdTomato-positive L cones (*Figure 5B*) or in the percentage of L cones with significant GFP expression (mean ± s.d.: 2.6% ± 1.8%%) (*Figure 5C*).

These results suggest that Tbx2a, but not Tbx2b, is important to preserve L cone identity. Without Tbx2a, L cones are unable to suppress M-opsin expression (*Sandkam et al., 2020*). Overall, analysis of *tbx2a* F0 mutants revealed that Tbx2a is important for the generation of UV cones and for maintaining L-cone identity.

## Tbx2b inhibits M-opsin expression in S cones

After identifying an additional role for Tbx2a, we turned our analysis to Tbx2b. Our RNA-seq revealed that in addition to UV cones, *tbx2b* is expressed in S cones (*Figure 4A*). Furthermore, our qPCR quantification showed an increase in M-opsin expression in *tbx2b* F0 mutants — specifically of *opn1mw1* and *opn1mw2* (*Figure 4—figure supplement 1*) — and in S-opsin expression. Based on these results, we tested whether Tbx2b is involved in control of S- or M-cone identity.

For our analysis of S and M cones in *tbx2b* F0 mutant larvae, we used the M-opsin reporter line — *Tg(opn1mws2:GFP)* — in combination with an S-cone reporter line — *Tg(opn1sw2:nfsB-mCherry)*. In control larvae, expression of the reporter proteins is largely non-overlapping, except for a small fraction of S cones that consistently express GFP (*Figure 6A*, top; *Tsujimura et al., 2007*). In *tbx2b* F0 mutants, we did not find significant changes in the number of S cones (identified by mCherry expression) (*Figure 6B*), but as described above, we did observe a clear increase in the number of GFP-positive cells (presumptive M cones). Furthermore, in *tbx2b* F0 mutants, we found that this increase in GFP expression was restricted to S cones, which become double-positive for GFP and mCherry expression (*Figure 6A*, bottom). We quantified the fraction of mCherry-positive S cones with GFP expression, and found that, in control larvae, this fraction is low (mean ± s.d.: 4.6% ± 7.9%). In comparison, in *tbx2b* F0 mutants this fraction is significantly higher (mean ± s.d.: 54.2% ± 26.9%) (*Figure 6C*). This abnormal increase in GFP expression in S cones in *tbx2b* F0 mutants, combined with the increase in M-opsin expression found in our qPCR analysis, indicates a loss of inhibitory control over the M-opsin promoter. By manually excluding double-positive cells, we counted GFP-only cells to quantify M cone densities and found that no significant changes in *tbx2b* F0 mutants (*Figure 6—figure supplement 1A–B*).

As a control, we repeated this S- and M-cone assessment in *tbx2a* F0 mutants. We again observed an increase in GFP-positive cells in *tbx2a* F0 mutants but without any significant changes in the number of S cones (*Figure 6B*) or in the fraction of mCherry-positive S cones with significant GFP expression (mean ± s.d.: 3.9% ± 3.5%) (*Figure 6C*). These results again corroborate our RNA-seq data showing that, while Tbx2a is not expressed by S cones and is not involved in the generation of S-cones, expression of Tbx2b in S cones is important to maintain their identity. Without Tbx2b, S cones are unable to suppress the expression of M-opsin. Overall, analysis of *tbx2b* F0 mutants revealed that Tbx2b is important for the generation of UV cones and for maintaining S-cone identity.

In summary, our results suggest that Tbx2 paralogs are required for the generation of UV cones and critical for establishing and maintaining the distinct identities of L cones and S cones. The discovery of these new roles of Tbx2 in photoreceptors demonstrates the power of the methods and techniques presented in this study.

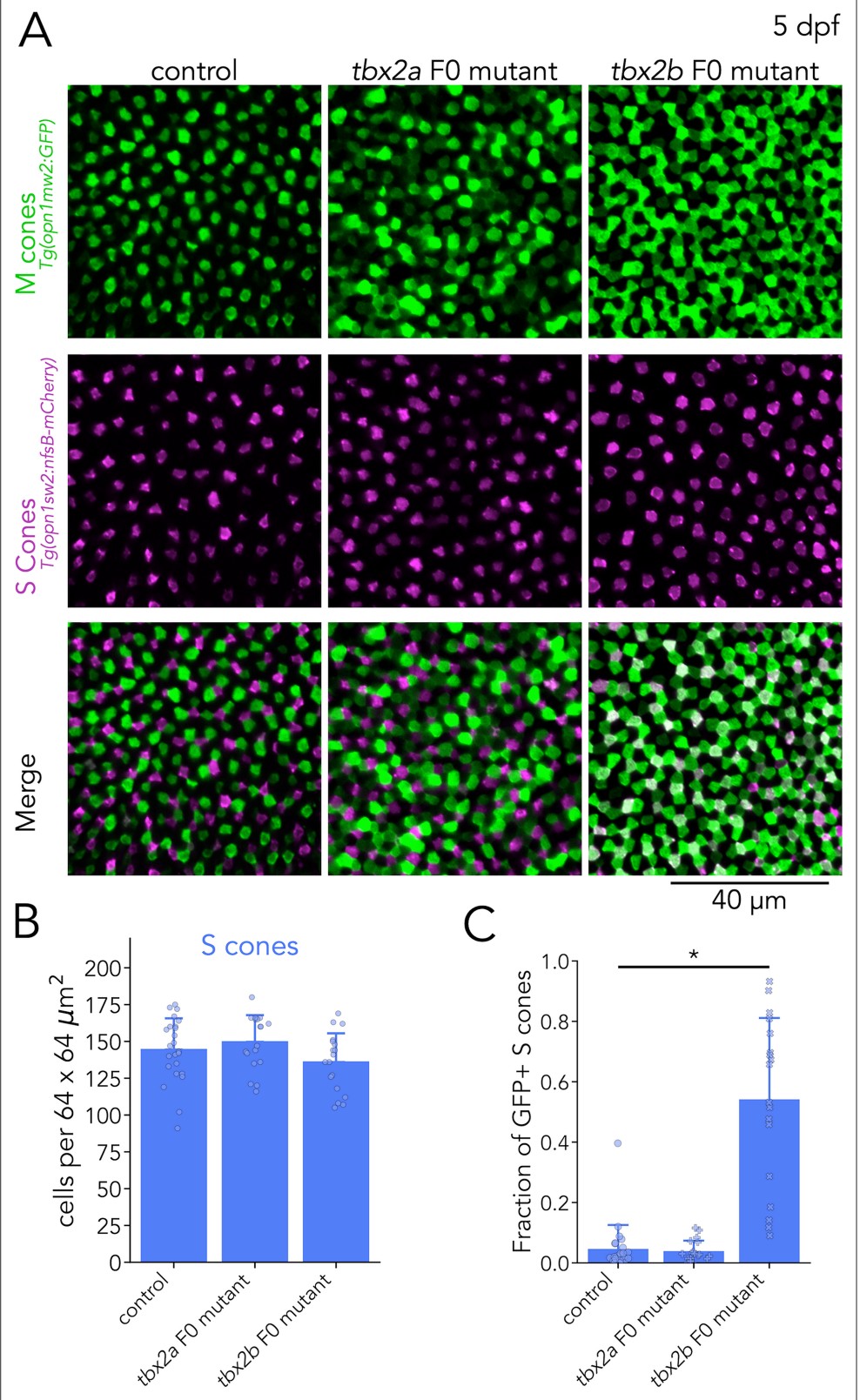

**Figure 6.** Tbx2b inhibits M-opsin expression in S cones. (**A**) Representative confocal images of the central retina of control, *tbx2a* and *tbx2b* F0 mutants at 5 dpf, in double transgenic larvae that label M cones — or M-opsin expressing cells — with GFP (green) and S cones with mCherry (magenta). Both *tbx2a* and *tbx2b* F0 mutants display an increase in GFP-positive cells. In *tbx2a* F0 mutants, increase in GFP signal is excluded from mCherry-

*Figure 6 continued on next page*

*Figure 6 continued*

positive cells, producing a decrease in the space without fluorescence, while in *tbx2b* F0 mutants, increase in GFP signal is restricted to mCherry-positive cells, which appear as double positive (white) in the merged images. (**B**) Quantification of S cones in the central retina shows no significant changes in either *tbx2a* or *tbx2b* F0 mutants compared to control (Kruskal-Wallis H=3.668, p=0.16, $n_{wt}$ = 24, $n_{tbx2a}$=18, $n_{tbx2b}$=18). (**C**) Quantification of the fraction of GFP-positive S cones (double positive cells in A) reveals a significant increase only in *tbx2b* F0 mutants (Kruskal-Wallis H=35.584, p=$1.87 \times 10^{-8}$, $n_{wt}$ = 24, $n_{tbx2a}$=18, $n_{tbx2b}$=18; Conover-Iman *posthoc* corrected p-values: control *vs.* *tbx2a* p=$2.87 \times 10^{-5}$, control *vs.* *tbx2b* p=0.63, *tbx2a* *vs.* *tbx2b* p=$1.13 \times 10^{-6}$).

The online version of this article includes the following figure supplement(s) for figure 6:

**Figure supplement 1.** Assessment of M-cone density in *tbx2a* and *tbx2b* F0 mutants.

## Discussion

The tools, resources and methods presented here provide a path to accelerate discovery in photoreceptor and retinal biology, to understand how photoreceptors acquire and maintain their final identities and subtype-specific specializations (e.g. morphology, phototransduction, metabolism, synaptic wiring, *etc*.). We have generated transcriptomic profiles from photoreceptors with unmatched depth and purity. These transcriptomes can be used to explore previously unrecognized gene-expression patterns across photoreceptor subtypes. Importantly, we can reliably assess if transcription factors play a role in controlling fate decisions between subtypes during photoreceptor development. We also demonstrate how F0 screening can be applied as a rapid, efficient, and flexible platform to create and study loss-of-function mutations. In our study, we apply F0 screening to investigate eight different transcription factors and test their involvement in photoreceptor development. Together, these methods provide an excellent in vivo setting to discover the function of other novel genes identified in our RNA-seq dataset. To facilitate future studies, we provide open and easy access to our transcriptomic dataset and analysis, and integrate it with other relevant and available datasets (https://github.com/angueyraLab/drRNAseq/). Ultimately, the knowledge gained by exploring these datasets can be used to inform strategies to control the photoreceptor differentiation in organoids — a potential gateway for cell-replacement therapies in retinal degenerations.

### Relation to other transcriptomic datasets

Recent studies have derived transcriptomes from zebrafish retinal cells and contain information from adult photoreceptors that provide an excellent resource to benchmark the quality of transcriptomes presented here. In our study, we derived samples using manual collection for a cell-type specific, SMART-seq2-based approach (***Kunze, 2017***). Three other recent studies used a variety of methods to segregate cell types in the retina. In ***Sun et al., 2018***, rod transcriptomes were obtained by fluorescent-activated cell sorting (FACS) (***Sun et al., 2018***). In ***Hoang et al., 2020***, retinal-cell transcriptomes were obtained using a single-cell droplet-based (dropSeq) approach in adults and at several time points during development (***Hoang et al., 2020***). Finally, in Ogawa and Corbo (2021), transcriptomes from adult zebrafish photoreceptors were obtained by enrichment through FACS followed by dropSeq (***Ogawa and Corbo, 2021a***).

We find that there are general consistencies across these datasets, which can be exemplified by focusing on phototransduction genes: we identify rod-enrichment in 26 of 27 phototransduction genes that are known to be rod-specific, while ***Sun et al., 2018*** identify 22 and ***Ogawa and Corbo, 2021a*** identify 23. We identify cone-enrichment in 31 of 35 phototransduction genes known to be cone specific, with high similarity to the subtype-specific expression patterns of ***Ogawa and Corbo, 2021a***. We found that these expression patterns are obscured in ***Hoang et al., 2020*** due to contamination with rod transcripts in all the retinal cells derived from adults — many known rod-specific genes are present in all photoreceptor subtypes (***Figure 1—figure supplement 3A***). Rods are the predominant cell type in the zebrafish adult retina — constituting ~40% of all photoreceptors (***Fadool, 2003***). In our experience, rods are fragile during dissociation and rod contamination presents a challenge to obtaining pure, subtype-specific datasets. Rhodopsin (*rho*) detection in non-rod samples is a simple way to assess contamination. We find that samples in ***Sun et al., 2018*** and in ***Hoang et al., 2020*** have significant rod contamination (>15%), while in ***Ogawa and Corbo, 2021a*** and in the data presented here, the rod contamination is low (<5%) (***Figure 1—figure supplement 3B***). Transcriptome depth

was considerably higher in our study compared to all other datasets. (*Figure 1—figure supplement 3C*). The high signal-to-noise ratio in our transcriptomes allows the detection of significantly more differentially-expressed genes (DEGs). In *Ogawa and Corbo, 2021a*, the authors detect 805 DEGs between photoreceptor subtypes (their report of ~1100 DEGs includes those that differentiate bipolar cells from photoreceptors). In our dataset, with more stringent criteria, we identify 3058 unique DEGs (Supplementary Data 1); 598 genes are shared by both datasets, 207 are unique to *Ogawa and Corbo, 2021a*, and 2460 are unique to this study. This higher signal-to-noise ratio is apparent for the targets of our F0-screen — *nr2e3, foxq2, skor1a, sall1a, lrrfip1a, xbp1, tbx2a,* and *tbx2b*. In particular, the restricted expression of *tbx2a* in UV and L cones — confirmed by our F0-screen results — is only apparent in our dataset (*Figure 1—figure supplement 3D*).

Overall, we find that the methods presented in this study are especially useful to generate high-quality transcriptomes of targeted cells. High depth and low contamination increase the statistical confidence and allow the detection of genes expressed at relatively low levels (e.g. *tbx2a* expression in L cones). Our method nicely complements dropSeq approaches that sample many more cells, which is especially advantageous for discovering new cell types or tracking developmental trajectories. For example, the dropSeq datasets find clear transcriptional differences between *opn1mw4*-expressing M cones and other M cones; we were not able to assess such differences due to our choice of using an M-cone reporter line — *Tg*(*opn1mw2:GFP*) — that does not label these M cones. In our view, these techniques are complementary and integration across datasets is imperative. To facilitate such comparisons, we have created an interactive plotter that integrates analysis across the datasets as outlined here. This resource is openly available and allows easy exploration and direct comparisons across datasets (https://github.com/angueyraLab/drRNAseq/), and includes the code and data needed to replicate our analyses. The expression plots presented here for all studies can be directly generated in this interactive plotter.

## Reliability and efficiency of F0 screening

The generation of loss-of-function mutants remains a cornerstone to test gene function. We use an F0 screen to accelerate the discovery of genes involved in establishing specializations between photoreceptor subtypes. Overall, we were able to create mutations in targeted genes in more than 80% of injected larvae by using 2–3 gRNAs (*Table 2*), and we were able to reliably phenocopy germline mutants, as exemplified by the loss of S cones in *foxq2* F0 mutants and the loss of rods in *nr2e3* F0 mutants. Interestingly, we find that *nr2e3* F0 mutants also have a decrease in UV-cone density. We speculate that Nr2e3, which is expressed transiently by early cone progenitors (*Figure 2—figure supplement 2F*; *Hoang et al., 2020*; *Xu et al., 2020*; *Haider et al., 2006*; *Alvarez-Delfin et al., 2009*), may play a role in the survival of developing UV cones, which we will pursue in the future. These findings highlight the flexibility of this screening method.

## Tbx2 and the diversification of photoreceptor subtypes

After the success of uncovering phenotypes in *foxq2* and *nr2e3* F0 mutants, we explored the effects of *tbx2* mutations in photoreceptor diversification. Our analyses revealed that Tbx2 is connected to properly generate all photoreceptor subtypes in zebrafish.

First, we showed that *tbx2a* and *tbx2b* are both expressed in UV cones, and the loss of either gene impairs the generation of UV cones. The high conservation in the amino acid sequence of TBX2 across vertebrates and the specific expression in evolutionarily related cone subtypes (*opn1sw1*-expressing photoreceptors in zebrafish, chicken, squirrel and primate) (*Yamagata et al., 2021*; *Kunze, 2017*; *Peng et al., 2019*) suggests that TBX2 may play a similar role across vertebrate species. We find that loss of UV cones in either *tbx2a* F0 mutants or *tbx2b* F0 mutants is associated with an increase in the number of rods during development. The switch in fate from UV cone to rod suggests that Tbx2a and Tbx2b play a role in an early fate decision in photoreceptor progenitors, allowing the acquisition of UV-cones by actively repressing rod fate. Interestingly the increase in rods (or rhodopsin expression) was not equal between *tbx2a* and *tbx2b* F0 mutants, suggesting that the two transcription factors regulate downstream targets differently. In addition, in vitro experiments have shown that Tbx2 binds to DNA as a monomer (*Sinha et al., 2000*), which makes the possibility of Tbx2a/Tbx2b dimers unlikely. Currently, it remains unclear why the generation of UV cones in zebrafish would require both paralogs.

Second, we show that *tbx2a* and *tbx2b* are expressed in L cones and S cones, respectively. Further, Tbx2a and Tbx2b help maintain L-cone and S-cone identity by repressing the expression of M opsin in vivo. A recent study in cichlids demonstrated that Tbx2a can bind and directly regulate the M-opsin promoter in vitro (*Sandkam et al., 2020*). This work also found that expression of *tbx2a* correlated strongly with the relative expression of M and L opsins, which cichlid species use to adjust their overall spectral sensitivity and match the requirements imposed by their habitats. Our work mainly focuses on opsin expression as a readout of photoreceptor identity. In the future, it will be interesting to investigate what additional changes in gene expression accompany changes in photoreceptor identity. Importantly, our findings highlight that Tbx2 not only plays a role in UV cone generation but is also important to maintain the identity of L and S cones.

Finally, our nuclear quantifications suggest that the loss of UV cones in *tbx2a* and *tbx2b* mutants is followed, not only by an increase in rods, but also by partial compensation by other cones. Surprisingly, we do not find an increase in the density of any of the other cone subtypes that could explain this attenuated loss in total cone density, as assessed using the transgenic reporter lines. It is likely that the unexplained gap between these quantifications corresponds to cones not labeled by our transgenic lines. In the future, it will be important to ascertain the identity of these cells, using germline mutants and other markers beyond opsins that define each subtype.

## Outlook

While conducting the experiments described in this paper, we learned a few lessons worth highlighting. First, we find that manual picking targeted cell types allowed us to focus on collecting healthy cells and generate transcriptomes of high depth and quality. An important advantage of this method is that barriers imposed by a cell type with a low density can be largely ignored if the targeted cell types can be recognized. For this reason, we think it would be interesting to apply this technique to fully understand further subdivisions of each photoreceptor subtype including the differences between *opn1lw1*- and *opn1lw2*-expressing L cones (*Mackin et al., 2019*) or between *opn1mw4*-expressing M cones and other M cones in zebrafish (*Ogawa and Corbo, 2021a*). Furthermore, it would be useful to explore regional specializations across the retina like the one proposed for UV cones in the acute zone (*Yoshimatsu et al., 2020*), and for fovea *vs.* periphery differences in primates (*Peng et al., 2019*). This manual-picking technique is likely to also be useful beyond photoreceptors to dissect differences between subtypes of other retinal cells.

Second, we find it is critical to create fast and easy access to multiple transcriptomic datasets. Eliminating technological barriers is important to ensure data can be accessed by all users. By ensuring proper access, new hypotheses pertaining to factors involved in photoreceptor development and other aspects of photoreceptor biology can be more readily explored. For example, many orthologs of human genes associated with retinal degenerations show high expression in zebrafish photoreceptors. For these reasons, we have taken a special effort to provide an interactive plotter that allows open exploration of four RNA-seq datasets in a single place. We expect that this tool will be valuable to the scientific community.

Third, the results of our F0 screen highlight some important features of how photoreceptors acquire their final identity. The process of specification seems to require several stages: defects in early stages can lead to a loss of subtypes (e.g. S cones in *foxq2* mutants) or to a change in identity (e.g. rods and UV cones in *tbx2* mutants), while defects in later stages can lead to alterations in identity without a loss of subtypes (e.g. misexpression of M opsin in *tbx2* mutants). In addition, while some transcription factors (like Foxq2 but also Thrb) mainly play a role in activating a particular fate, others (like Tbx2, but also Nr2e3 or Prdm1) play a role in inhibiting the fate of other cell types (*Swaroop et al., 2010*; *Brzezinski et al., 2010*). Because of its conserved sequence and expression, TBX2 may play a similar role in mammalian S cones — actively repressing the fate of rods. Such active repression is most likely a fundamental mechanism to maintain subtype identity throughout the life span of an organism. These mechanisms of cell identity echo beyond photoreceptors into the context of the generation of any cell subtype. In fact, TBX2 plays a similar repressive role in the inner ear of mice (*García-Añoveros et al., 2022*).

Our current study focused on the differential expression of transcription factors because of their central role in subtype diversity. A similar approach to the one outlined here can be used to study the function of genes involved in phototransduction, metabolism, ciliary transport, synaptic machinery,

*etc.* It is likely that the other targets of our F0 screen — *skor1a*, *sall1a*, *lrrfip1a* and *xbp1* — that have no clear involvement in the generation of photoreceptor subtypes, may play a role in regulating these other aspects of photoreceptor biology. The dataset and methods described here are an excellent resource to propose hypotheses, to generate an initial list of candidate genes and to perform efficient screening for phenotypes related to these other functions.

# Materials and methods

## Key resources table

| Reagent type (species) or resource | Designation | Source or reference | Identifiers | Additional information |
|---|---|---|---|---|
| Gene (*Danio rerio*) | GRCz11 | GenBank | RefSeq: GCF_000002035.6 GenBank: GCA_000002035.4 | |
| Strain, strain background (*Danio rerio*) | Tg(xOPS:GFP)<sup>fl1Tg</sup> | *Fadool, 2003* | ZFIN:ZDB-ALT-080517–1 | |
| Strain, strain background (*Danio rerio*) | Tg(–5.5opn1sw1:GFP)<sup>kj9Tg</sup> | *Takechi et al., 2003* | ZFIN: ZDB-ALT-080227–1 | |
| Strain, strain background (*Danio rerio*) | Tg(opn1sw1:nfsB-mCherry)<sup>q28Tg</sup> | *Yoshimatsu et al., 2014* | ZFIN:ZDB-ALT-160425–1 | |
| Strain, strain background (*Danio rerio*) | Tg(–3.5opn1sw2:GFP)<sup>kj11Tg</sup> | *Takechi et al., 2008* | ZFIN: ZDB-ALT-090622–2 | |
| Strain, strain background (*Danio rerio*) | Tg(opn1sw2:nfsB-mCherry)<sup>q30Tg</sup> | *Yoshimatsu et al., 2014* | ZFIN: ZDB-ALT-160425–3 | |
| Strain, strain background (*Danio rerio*) | Tg(opn1mw2:GFP)<sup>kj4Tg</sup> | *Tsujimura et al., 2007* | ZFIN: ZDB-ALT-071206–2 | |
| Strain, strain background (*Danio rerio*) | Tg(thrb:tdTomato)<sup>q22Tg</sup> | *Suzuki et al., 2013* | ZFIN: ZDB-ALT-131118–3 | |
| Antibody | Anti-zebrafish S opsin (Rabbit polyclonal) | Kerafast | Cat# EJH012 | Immunolabelign (1:200) |
| Antibody | anti-rhodopsin [1D4] (Mouse monoclonal) | Abcam | Cat# ab5417 [1D4] | Labels L opsin in zebrafish Immunolabeling (1:200) |
| Software, algorithm | HiSat2 | *Kim et al., 2019* | | |
| Software, algorithm | Stringtie | *Pertea et al., 2016* | | |
| Software, algorithm | DeSeq2 | *Love et al., 2014* | | |
| Software, algorithm | Seurat | *Satija et al., 2015* | Seurat v4.0 | https://satijalab.org/seurat/ |
| Software, algorithm | R/Rstudio | *R Development Core Team, 2021*; *RStudio, 2020* | R version 4.2.0 | https://www.r-project.org/ https://posit.co/ |
| Software, algorithm | Python | *https://www.python.org/* | Python 3.10 | |
| Software, algorithm | JupyterLab | *https://jupyter.org/* | JupterLab version 3.2.1 | |
| Software, algorithm | Napari | contributors, 2019 | Napari0.4.17 | *https://napari.org/* |
| Software, algorithm | Cellpose | *Stringer et al., 2021* | Cellpose2.0 | *https://www.cellpose.org/* |
| Software, algorithm | FIJI/ImageJ | *Schindelin et al., 2012* | FIJI | *https://imagej.net/software/fiji/* |

## Animals

We grew zebrafish larvae at 28 °C in E3 embryo media (5 mM NaCl, 0.17 mM KCl, 0.33 mM CaCl2, and 0.33 mM MgSO4, buffered in HEPES, pH = 7.2) under a 14 hr:10 hr light-dark cycle (lights on from 8 A.M. to 10 P.M.). At 1 dpf, we added 0.003% 1-phenyl-2-thiourea (PTU) to the embryo medium to block melanogenesis. All work performed at the National Institutes of Health was approved by the NIH Animal Use Committee under animal study protocol #1362–13. For RNA-seq samples with adult zebrafish, animals of both sexes were used. For the F0-screen, larvae were examined at 5 dpf. At these

age, sex cannot be predicted or determined, and therefore the sex of the animals was not considered. The transgenic lines used in this study are listed in *Table 1*.

## RNA-seq sample collection

We euthanized adult zebrafish by immersion in ice-cold water (below 4 °C) followed by decapitation. To avoid influences of circadian changes in gene expression, we collected all samples between 3 and 6 hr after light onset (11 A.M. – 2 P.M.), the period of highest sensitivity to visual stimuli (*Li and Dowling, 1998*). We pierced the cornea with a 30-gauge needle and removed the cornea and lens before performing enucleation. Once the eye was isolated, we gently separated the retina from sclera and RPE using fine forceps or electrically-sharpened tungsten electrodes (*Protocols, 2012*) and immediately started incubation in papain solution (5 U/mL papain Calbiochem#5125, 5.5 mM L-Cysteine, 1 mM EDTA in divalent-free Hank's balanced salt solution) for 35 min at 28 °C. After a brief wash in DMEM supplemented with 5% bovine serum albumin, we performed mechanical trituration of the retina with the tip of a 1 mL pipette and used a cell-strainer polystyrene tube to obtain a single-cell suspension. After spin-down (2000x G for 2 min), we resuspended cells in 500 µL of enzyme-free fresh DMEM and diluted the cell suspension into three serial 10-fold dilutions before plating in glass-bottom petri dishes. The dilutions ensured that we could find a preparation where the density of cells and debris was low, and most photoreceptors were truly isolated. We inspected the cell suspension using an epifluorescence microscope (Invitrogen EVOS cell-imaging system) and, for each sample, we collected and pooled 20 photoreceptors per retina based on their fluorescence and morphology (prioritizing cells that looked healthy, had intact outer segments, visible mitochondrial bundles, and undamaged cell membranes) using an oil-based microinjector system (Eppendorf CellTram 4 R) and glass pipettes with a 15 µm opening (Eppendorf TransferTip-ES). After collection, we resuspended photoreceptors in 1 µL of fresh PBS, reinspected cells for fluorescence, collected them in a PCR tube containing 8 µL of lysis buffer of the RNA kit and kept the tube on ice until cDNA libraries were prepared. We used the SMART-seq v4 ultra-low input RNA kit for sequencing (Takara #634897) using the manufacturer's instructions for single-cell samples, followed by the Low Input Library Prep Kit v2 (Takara #634899). For sequencing, we pooled up to 12 samples (with different barcodes) in one lane of a flow cell (Illumina HiSeq 2500) and used a 150 bp paired-end read configuration. The first sequencing batch contained 4 UV-cone and 4 S-cone samples in a single flow cell, and the second sequencing batch contained the rest of the samples divided across 2 flow cells (6 rod, 1 UV-cone, 2 S-cones, 7 M-cones and 6 L-cone samples). In summary, each sample consisted of a pool of 20 photoreceptors of a single subtype and our analysis relied on 5 biological replicates for UV cones, 7 for M cones and 6 each for rods, S cones and L cones.

## RNA-seq data analysis

After an initial quality control and trimming of primer and adapters sequences using Trimmomatic (*Bolger et al., 2014*), we used the NIH high-performance computing resources (Biowulf) to align reads to the zebrafish genome (*Danio rerio GRCz11*) using HiSat2 (*Kim et al., 2019*) and to assemble and quantify transcripts using Stringtie (*Pertea et al., 2016*). We performed differential expression analysis using Deseq2 and pcaExplorer for initial visualizations (*Love et al., 2014*; *Marini and Binder, 2019*). Genes were considered as differentially expressed if fold-enrichment >1.5, p-value <0.01 and the estimated false positive rate or p-adjusted <0.1. In addition, genes were required to have positive reads in >50% of the enriched samples. To be able to detect differences that relied on expression on just 1 or 2 cone subtypes, we removed the requirement on fold-enrichment in rod *vs.* cone comparisons. To further explore the data, we transformed read numbers into fragments per kilobase per million reads (FPKM) (Supplementary Data 01) and developed custom routines in Python for plotting. We subselected transcription factors by selecting genes identified with 'DNA-binding transcription factor activity' in ZFIN (*Bradford et al., 2021*) and repeated principal component and differential expression analyses (Supplementary Data 02 and 03). Transcription factors were considered as significantly expressed if at least 20% (i.e. 7 out of 35) of the samples had positive reads. To ensure broad access to our transcriptomic data, we provide access to the raw data (GEO accession number GSE188560), and after analysis in several formats including as a plain csv file, as a Seurat object for easy integration with dropSeq datasets (*Satija et al., 2015*), and finally, as an interactive database for easy browsing and visualization (*https://angueyraLab.github.io/drRNAseq/lab*). To make direct comparisons between

our data and other RNA-seq studies, we have integrated visualizations that use their publicly available data. For rod transcriptomes obtained using FACS, we used the provided analyzed data, which includes gene log counts per million (cpm) for four rod samples (GFP-positive cells) and four non-rod samples (GFP-negative retinal cells) (*GSE100062*) (*Sun et al., 2018*). For transcriptomes from adult photoreceptors obtained using FACS followed by dropSeq (*Ogawa and Corbo, 2021a*), we used the Seurat object provided by the authors (*GSE175929*) and we used custom scripts in R (*R Development Core Team, 2021*), using Rstudio (*RStudio, 2020*) and Seurat (*Satija et al., 2015*) to export average expression values and percent of cells with positive counts of each gene for each cluster. For transcriptomes of retinal cells obtained using dropSeq (*Hoang et al., 2020*), we used the Seurat object for zebrafish development provided by the authors (http://bioinfo.wilmer.jhu.edu/jiewang/scRNAseq/), and we updated the object to Seurat v03 (*Stuart et al., 2019*), extracted cells that corresponded to adult rods and cones, performed clustering and used the expression of opsins and other markers to identify cone subtypes (including *arr3a* for L and M cones, *arr3b* and *tbx2b* for UV and S cones, *thrb* and *si:busm1-57f23.1* for L cones and *foxq2* for S cones). All results and scripts necessary to recreate these analyses are also provided openly (https://github.com/angueyraLab/drRNAseq). We have also included analysis on developing photoreceptors using this dataset, to replicate results presented in *Figure 2—figure supplement 2*.

## Fluorescent in situ hybridizations (RNAscope)

We performed the RNAscope assay following manufacturer's instructions (ACDBio) for fresh frozen samples, with the following custom-made probes: negative control (T1-T12), *actb2*-T2, *gnat2*-T3, *foxq2*-T2, *tbx2a*-T3, *skor1a*-T6, *lrrfip1a*-T7, *cux2b*-T10, *smad5*-T11, *ahr1b*-T2, *etv5a*-T3. After euthanasia, we collected eyes from adult zebrafish, embedded them in plastic molds filled with cryo-embedding medium (OCT) and froze immediately at –80 °C. We obtained 15 µm cryo-sections and stored them at –80 °C until use. We fixed retinal sections by immersion in 4% paraformaldehyde for 60 min, performed washes with RNase-free PBST (PBS + 0.01% Tween) and dehydration in methanol in a step-wise manner (5 min incubation each in 25%, 50%, 75%, and 100% methanol in PBST), before air drying for 5 min. We then applied Protease III for 5 min at room temperature, performed three washes with PBST, and hybridization with probes for 2 hr at 40 °C in a humidified tray. After hybridization, we interleaved three 5-min washes (with the provided Wash Buffer) and incubation with Amp1, Amp2, Amp3 solutions for 30 min, and with the Fluoro solution for 15 min at 40 °C in a humidified tray. After the final washes, the sections were immediately covered with mounting medium and a coverslip before imaging. To combine this assay with reporter lines, we omitted the protease treatment, but this led to a decrease in probe staining. Decreasing Protease treatment from the recommended 30 minutes to 5 minutes improved the morphology of the tissue but did not preserve GFP fluorescence.

## F0-CRISPR screening

We designed guide RNAs (gRNAs) using the online resource *CHOPCHOP* (*Labun et al., 2021*). We selected guides that targeted exons that encode the DNA-binding domains of transcription factors, had no self-complementarity, and that had 3 or more mismatches with other regions of the zebrafish genome; if this was not possible, we targeted the first coding exon (*Table 2*). We used purified Cas9 protein (Alt-R S.p. Cas9 nuclease, v.3) and chemically synthesized AltR-modified crRNA and tracrRNA (Integrated DNA technologies) for injections (*Hoshijima et al., 2019*; *Table 2*). We prepared 1 µL aliquots of a 25 µM stock solution of Cas9 protein diluted in 20 mM HEPES-NaOH (pH 7.5), 350 mM KCl and 20% glycerol, and stored them at –80 °C until use. We diluted each target-specific crRNA and the common tracrRNA using the provided duplex buffer as a 100 µM stock solution and stored them at –20 °C. We prepared a 50 µM crRNA:tracrRNA duplex solution by mixing equal volumes of the stock solutions followed by annealing in a PCR machine (95 °C, 5 min; cooling 0.1 °C /s to 25 °C; 25 °C for 5 min; rapid cooling to 4 °C), then we used the duplex buffer to obtain a 25 µM stock solution, before mixing equal volumes of the guides targeted to a single gene (3 guides for *skor1a*, *sall1a*, *xbp1*, *tbx2a* and *tbx2b*, 2 guides for *foxq2* and *nr2e3*), making aliquots (2 µL for *foxq2* and *nr2e3*, 3 µL for the other genes) and storing at –20 °C until use. Prior to microinjection, we prepared 5 µM RNP complex solutions by mixing 1 µL of 25 µM Cas9, 1 µL of 0.25% phenol red and 3 µL of the *tbx2a* or *tbx2b* duplex solution, or 1 µL of pure water and 2 µL of the duplex solution for the other genes. We incubated the RNP solution at 37 °C for 5 min and kept at room temperature for use in the following

**Table 3.** Primer sequences for genotyping.

| Gene | Primer (5' — 3') | Product size |
|---|---|---|
| *foxq2* F | TGCTCTTCAAACAGGACAAGAA | |
| *foxq2* R | TTCCAGCACATGCAGAAATAAT | 406 bp |
| *nr2e3* F | TTCAGACAGCATAGGGTGACAT | |
| *nr2e3* R | CTCACCTGTAGATGAGTCTGCG | 253 bp |
| *tbx2a* F | CGTTCATTCGAATTCATTGTGT | |
| *tbx2a* R | TGTTTTGATGTCGCTGATTTTC | 462 bp |
| *tbx2b* F | TGACGAGCACTAATGTCTTCCT | |
| *tbx2b* R | GCATCGCAGAACGAAAGTAGAT | 309 bp |
| *skor1a* F | CTACAACGAAATTCACAACCGA | |
| *skor1a* R | GCGGTGCGAATGAAATATAAA | 349 bp |
| *xbp1* F | ATTTCCCACCCCTAATCAAAAC | |
| *xbp1* R | GGCTCAGATGTGTGAGTCTCTG | 269 bp |
| *sall1a* F | ATACTTGACAAAGAGGAGGCCA | |
| *sall1a* R | TGAGGTAGTGAGGCAGAGATGA | 179 bp |
| *lrrfip1a* F | CGATTCCACTTCCTCAATTGTT | |
| *lrrfip1a* R | AGCACACTGCCTGAATAAAACAT | 281 bp |

2–3 hr. We injected ~1 nL of the 5 µM RNP complex solution into the cytoplasm of one-cell stage zebrafish embryos.

## Genotyping

We extracted DNA from the bodies of larvae (5 dpf) after enucleation by placing them in 25 µL of 25 mM NaOH with 0.2 mM EDTA, heating to 95 °C for 30 min, and cooling to 4 °C. Then we neutralized the solution by adding 25 µL of 40 mM Tris-HCl and vortexed the samples. For genotyping, we used a fluorescent PCR method (*Carrington et al., 2015*). We added the M13F adapter sequence (5'-TGTAAAACGACGGCCAGT-3') to forward primers and the PIG-tail adapter sequence (5'-GTGT CTT-3') to reverse primers and used incorporation of fluorescent M13F-6FAM for detection. Our PCR mixture (1 x), for a 20 µL reaction, contained forward primer (0.158 µM), reverse primer (0.316 µM), M13-FAM (0.316 µM, IDT), Phusion HF PCR Master Mix (1 x, BioLabs), water (6.42 µL), and 2 µL of DNA. We used the following PCR protocol: (1) 98 °C denaturation for 30 s, (2) 34 cycles of 98 °C for 10 s, 64–67°C for 20 s, 72 °C for 20 s (3) final extension at 72 °C for 10 min, (4) hold at 4 °C. All primers and expected sizes are provided in *Table 3*, and the estimated efficiency of producing mutations with each guide combination in preliminary experiments is included in *Table 2*. Because of the high homology between *tbx2a* and *tbx2b*, we also tested cross-reactivity of the guides between these two genes and found no sign of mutations in the non-targeted gene (0/48 larvae tested).

## Quantitative PCR (qPCR)

We euthanized groups of 20–30 zebrafish 5 dpf larvae by immersion in ice-cold water (below 4 °C) and immediately performed RNA extraction using the RNeasy Mini Kit (Qiagen) and reverse transcription using the High-Capacity cDNA Reverse Transcription Kit (Thermo Fisher), which relies on random primers. Samples were kept frozen (–20 °C) until use. For qPCR assays, we used the PowerUp SYBR Green Master Mix (Thermo Fisher) and a 96-well system (CF96, Biorad) following manufacturer's protocols. We estimated expression levels using the relative standard curve method, using five serial standard dilutions of cDNA obtained from wild-type larvae. To calculate fold differences in gene expression, we normalized transcript levels to the levels of actin-b2 (*actb2*), and all measurements were repeated in triplicate. We performed statistical testing using Kruskal-Wallis tests on the three groups (control, *tbx2a* F0 mutants and *tbx2b* F0 mutants) with a p-value <0.05 required for

**Table 4.** qPCR primers.

| Gene | Primer (5' to 3') |
| --- | --- |
| rho F | TCCGAGACCACACAGCG |
| rho R | CTGCTTGTTCATGCAGATG |
| opn1sw1 F | ATGGTCCTTGGCTGTTCTGG |
| opn1sw1 R | CCTCGGGAATGTATCTGCTCC |
| opn1sw2 F | GGAGGAATGGTGAGTTTGTG |
| opn1sw2 R | GGTCTTGAAGGTAAAGTTCC |
| opn1mw1 F | CAGCCCAGCACAAGAAACTC |
| opn1mw1 R | AGAGCAACCTGACCTCCAAGT |
| opn1mw2 F | TTTTTGGCTGGTCCCGATACA |
| opn1mw2 R | CAGGAACGCAGAAATGACAGC |
| opn1mw3 F | TGCTTTCGCTGGGATTGGATT |
| opn1mw3 R | CCCTCTGGAATATACCTTGACCA |
| opn1mw4 F | CACGCTTTCGCAGGATGC |
| opn1mw4 R | CGGAATATACCTGGACCAAC |
| opn1lw1 F | CCCACACTGCATCTCGACAA |
| opn1lw1 R | AAGGTATTCCCCATCACTCCAA |
| opn1lw2 F | AGAGGGAAGAACTGGACTTTCAGA |
| opn1lw2 R | TTCAGAGGAGTTTTGCCTACATATGT |
| actb2 F | GTACCACCAGACAATACAGT |
| actb2 R | CTTCTTGGGTATGGAATCTTGC |

significance. Significant results were followed up with a *posthoc* Conover-Iman test with a Bonferroni adjustment of p-value (***Conover and Iman, 1979***). All primers used for qPCR are provided in ***Table 4***.

## Immunohistochemistry

We fixed zebrafish larvae at 5 dpf in 4% paraformaldehyde in phosphate buffered saline (PBS) for 1 hr at room temperature, followed by washes with 1% Triton X-100 PBS (3x10 min). We incubated larvae in primary antibodies diluted in 2% normal donkey serum (Jackson ImmunoResearch) and 1% Triton X-100 PBS for five days at 4 °C with continuous and gentle shaking. To label S cones, we used a rabbit polyclonal anti-blue opsin (Kerafast EJH012) in a 1:200 dilution; to label L cones, we used a mouse monoclonal anti-rhodopsin antibody in a 1:200 dilution (Abcam 1D4 ab5417) (***Yin et al., 2012***). After incubation with primary antibodies, we performed washes with 1% Triton X-100 PBS (3x15 min). We incubated larvae in donkey polyclonal secondary antibodies labeled with Cy5 (Jackson ImmunoResearch) in 1% Triton X-100 PBS overnight at 4 °C with continuous and gentle shaking and performed washes in 1% Triton X-100 PBS (3x15 min) before mounting.

## Imaging
### Sample preparation and image acquisition
For larval imaging, we enucleated eyes from fixed larvae using electrically-sharpened tungsten wires (***Protocols, 2012***). We placed isolated eyes on a coverslip and oriented photoreceptors closest to the coverslip before using a small drop of 1.0% low-melting point agarose to fix them in place. Upon solidification, we added a polyvinyl-based mounting medium (10% polyvinyl alcohol type II, 5% glycerol 25 mM, Tris buffer pH 8.7 and 0.5 µg/mL DAPI) and placed the coverslip on a glass slide, separated by a spacer (Grace Biolabs and/or duct tape) to avoid compression. We used the bodies of the larvae for genotyping and imaged the corresponding larval retinas using a Nikon A1R resonant-confocal

microscope with a 25 x, 1.10 NA water-immersion objective. We acquired z-stack images from a 64 µm x 64 µm square area of the central retina (dorsal to optic nerve) for photoreceptor quantification every 0.4–0.5 µm at a 1024x1,024 pixel resolution.

For in situ hybridizations of retinal sections, we used the same mounting medium and imaging system, but used a 60 x, 1.40 NA oil-immersion objective and acquired z-stack images from a 70 µm x 70 µm square area centered on the photoreceptor layer, every 0.25 µm at a 1024x1,024 pixel resolution. Images correspond to maximum-intensity projections of 2–4 µm in depth, after contrast adjustments to reject the autofluorescence of outer segments and highlight the bright fluorescent puncta.

## Image analysis
### Photoreceptor quantification
We imported confocal z-stacks of the central region of the retina (64 µm x 64 µm) into Napari (**contributors, 2019**). We created maximum intensity projections (MIPs) using a small subset of the z-stack (2–10 planes) that ensured that we captured all photoreceptor cells in the region into a single image. We then used the Napari plugin of Cellpose, a machine-learning-based segmentation algorithm, to segment photoreceptors in each image, using the *cyto2* model (**Stringer et al., 2021**). Finally, we manually corrected the segmentation to ensure all photoreceptors were properly counted. For quantification of cone nuclei, we performed manual counts of DAPI stains in the same images used for photoreceptor quantifications, using the inner displacement of UV-cone nuclei to recognize UV cones from S, M and L cones. We performed statistical comparisons for counts of each photoreceptor subtype between clutchmate wild-type (*wt*) controls and mutant (*F0*) larvae using Kruskal-Wallis tests, with a p-value <0.01 required for statistical significance.

### Identification of single and double-positive cells in *tbx2* mutants
The increase in the *Tg(opn1mws2:GFP)*+ cells in *tbx2a* F0 mutants and *tbx2b* F0 mutants larvae made segmentation of the green channel difficult and unreliable, as these additional cells did not conform to the normal spatial separation between M cones. For this reason, we used the more accurate segmentation of L cones and S cones using the red channel, when imaging *Tg(thrb:tdTomato)* or *Tg(opn1sw2:nfsB-mCherry)*, respectively, and used it to create masks for the green channel. We normalized the GFP signal across the whole image to span a 0–1 range (to be able to make comparison between images) and used a 10-pixel erosion (to avoid effects due to optical blurring during imaging of the GFP signal) before calculating the average normalized GFP signal contained within each S-cone or L-cone. By plotting the distribution of GFP signal in L cones, we were able to establish a threshold of 0.195 that was exceeded by only 5.2% of L cones in control larvae and used it to classify L cones as GFP + in both control and F0 larvae. In the original work that established the *Tg(opn1mws2:GFP)* line, it was noted that a subset of S cones in control larvae are GFP+ (**Tsujimura et al., 2007**). We were able to identify these cells using a GFP signal threshold of 0.275 (4.6% of control S cones), and again used this same threshold to quantify the fraction of GFP + S cones in both control and F0 larvae. Subsequently, to quantify M cone densities in these mutants, we performed manual counts of single positive cells ('GFP only') by excluding cells previously segmented as L or S cones.

## Statistical analyses
We performed statistical analyses and data plots using Python in Jupyter notebooks (**Kluyver et al., 2016**). Values of data and error bars in figures correspond to averages and standard deviations, and for statistical comparisons we used Kruskal-Wallis tests with a p-value <0.01 required for significance, unless stated otherwise. For statistical comparisons in *tbx2* mutants, we performed Kruskal-Wallis tests on the three groups (control, *tbx2a* F0 mutants and *tbx2b* F0 mutants), and significant results were followed up with a *posthoc* Conover-Iman test with a Bonferroni adjustment of p-value (**Conover and Iman, 1979**). Samples sizes, test values and significance levels are stated in the figure legends. No randomization, blinding, or masking was used for our animal studies, and all replicates are biological. For RNA-seq, we performed an initial sequencing run after collecting dissociated photoreceptors in squirrel (**Kunze, 2017**) and zebrafish and established that a minimum of four samples per subtype were required to establish reliable statistical significance in differential gene-expression analysis. For F0 screening, our initial experiments were aimed at replicating the loss of UV cones and the increase in rods reported for *tbx2b* mutants (**Alvarez-Delfin**

*et al., 2009*), and we established that a minimum of 6 injected larvae per group were needed to provide enough statistical power in photoreceptor quantifications in F0 larvae. Injected larvae that had normal (wild-type) genotypes — a sign that CRISPR mutagenesis was not successful — were excluded from analysis, so that quantifications rely solely on larvae with confirmed mutations in the targeted gene.

## Acknowledgements

This work was supported by the National Eye Institute Intramural Research Program (WL), the National Institute on Deafness and Other Communication Disorders Intramural Research Program (*1ZIADC000085-01*, KSK), and National Eye Institute Pathway to Independence Award (*K99EY030144-01*, JA). This work utilized the computational resources of the NIH HPC Biowulf cluster. (*http://hpc.nih.gov*). We would like to thank Jamie Sexton, Alisha Beirl and Katherine Pinter for all the animal care and technical support, John Ball, Elizabeth Cebul and other members of the Kindt Lab and the Li Lab for useful discussions, and Matthew Brooks, Linn Gieser and Anand Swaroop for sequencing services. We are very grateful to Rachel Wong, Takeshi Yoshimatsu, Ralph Nelson, James Fadool, Steven Leach, Brian Perkins and Xiangyun Wei for providing the transgenic zebrafish lines used in this study.

## Additional information

### Funding

| Funder | Grant reference number | Author |
|---|---|---|
| National Eye Institute | K99EY030144-01 | Juan M Angueyra |
| National Institute on Deafness and Other Communication Disorders | 1ZIADC000085-01 | Katie Kindt |
| National Eye Institute | IRP | Wei Li |

The funders had no role in study design, data collection and interpretation, or the decision to submit the work for publication.

### Author contributions

Juan M Angueyra, Conceptualization, Resources, Data curation, Software, Formal analysis, Supervision, Validation, Investigation, Visualization, Methodology, Writing - original draft, Project administration, Writing - review and editing; Vincent P Kunze, Conceptualization, Investigation, Methodology, Writing - review and editing; Laura K Patak, Formal analysis, Validation, Investigation, Visualization, Writing - original draft, Writing - review and editing; Hailey Kim, Investigation; Katie Kindt, Wei Li, Conceptualization, Supervision, Funding acquisition, Writing - review and editing

### Author ORCIDs

Juan M Angueyra ⬤ http://orcid.org/0000-0002-9217-3069
Vincent P Kunze ⬤ http://orcid.org/0000-0002-7869-9793
Katie Kindt ⬤ http://orcid.org/0000-0002-1065-8215
Wei Li ⬤ http://orcid.org/0000-0002-2897-649X

### Ethics

This study was performed in strict accordance with the recommendations in the Guide for the Care and Use of Laboratory Animals of the National Institutes of Health. All work performed at the National Institutes of Health was approved by the NIH Animal Use Committee under animal study protocol #1362-13.

### Decision letter and Author response

Decision letter https://doi.org/10.7554/eLife.81579.sa1
Author response https://doi.org/10.7554/eLife.81579.sa2

# Additional files

## Supplementary files

• Transparent reporting form

• Supplementary file 1. Differential gene expression in zebrafish photoreceptors. Collection of CSV files containing output of differential gene expression analysis using DeSEQ2, along relevant directions (rods vs. cones; (UV+S) vs. (M+L); M vs. L; UV vs. S) and including counts (in FPKM) for all detected genes.

• Supplementary file 2. Differential transcription factor expression in zebrafish photoreceptors: rods vs. cones. CSV file containing transcription factors with significant differential expression between rod and cone samples.

• Supplementary file 3. Differential transcription factor expression in zebrafish photoreceptors: cone subtypes. Collection of CSV files containing transcription factors with significant differential expression between cone subtypes.

• Supplementary file 4. Transcriptomic dataset of adult zebrafish photoreceptors for Seurat. Transcriptomic data from zebrafish adult photoreceptors, converted to .rds object, for direct use in Seurat v3.0.

## Data availability

Sequencing data have been deposited in GEO under accession code GSE188560. Exploration of the RNAseq dataset has been made openly available and easy to use by novice users in https://github.com/angueyraLab/drRNAseq/ (copy archived at swh:1:rev:0a8d7697ab6e42fdd40e9d-fb758658b21b85d0f1). This includes comparisons with other zebrafish photoreceptor datasets that are available. RNA-seq data has been exported into Seurat v3.0 format (Supplementary file 4) for integration into other RNA-seq data analysis pipelines.

The following previously published datasets were used:

| Author(s) | Year | Dataset title | Dataset URL | Database and Identifier |
|---|---|---|---|---|
| Sun C, Galicia C, Stenkamp DL | 2018 | Zebrafish retinal rod photoreceptors | https://www.ncbi.nlm.nih.gov/geo/query/acc.cgi?acc=GSM2670720 | NCBI Gene Expression Omnibus, GSM2670720 |
| Hoang T, Wang J, Boyd P, Wang F, Santiago C, Jiang L, Yoo S, Lahne M, Todd LJ, Jia M, Saez C, Keuthan C, Palazzo I, Squires N, Campbell WA, Rajaii F, Parayil T, Trinh V, Kim DW, Wang G, Campbell LJ, Ash J, Fischer AJ, Hyde DR, Qian J, Blackshaw S | 2020 | Comparative transcriptomic and epigenomic analysis identifies key regulators of injury response and neurogenic competence in retinal glia | https://www.ncbi.nlm.nih.gov/geo/query/acc.cgi?acc=GSE135406 | NCBI Gene Expression Omnibus, GSE135406 |
| Ogawa Y, Corbo JC | 2021 | Single-cell profiling of photoreceptor cells in adult zebrafish | https://www.ncbi.nlm.nih.gov/geo/query/acc.cgi?acc=GSE175929 | NCBI Gene Expression Omnibus, GSE175929 |

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
