## [Editor Report]

This manuscript offers a valuable transcriptomic data set of known types of adult zebrafish photoreceptors (rod and cones). The study identifies a large set of differentially expressed transcription factors, many of which still have an unidentified function in photoreceptors and offers to the scientific community an interactive plotter to compare the present data with recent and similar studies. Using CRISPR F0 screening, the study shows that the two tbx2 zebrafish paralogues are involved in photoreceptors specification beyond what is currently known. The study uses a solid methodology that could be applied to other retinal cell types or other tissues.

---

## [Decision Letter]

**Decision letter after peer review:**

Thank you for submitting your article "Identification of transcription factors involved in the specification of photoreceptor subtypes" for consideration by *eLife*. Your article has been reviewed by 3 peer reviewers, one of whom is a member of our Board of Reviewing Editors, and the evaluation has been overseen by Marianne Bronner as the Senior Editor. The following individual involved in the review of your submission has agreed to reveal their identity: Daisuke Kojima (Reviewer #3).

Essential revisions:

The following points raised by the reviewers need to be addressed.

1) The analysis is based on already differentiated cells and therefore many of the TF needed for photoreceptors specification might not be expressed. This needs to be considered in the title, introduction, and discussion.

2) Revise statistical analysis using different statistical tests such as Kruskal-Wallis or similar.

3) Revise the claim that tbx2 is a master regulatory gene of photoreceptor development because not supported by the data and is rather inappropriate.

4) Revise inconsistent results related to Figure 1-supplement 1.

5) Revise nomenclature for M opsin, FØ and the use of terms like two-factor-authentication.

6) Revise the discussion related to the F0 screening, which is rather long for a technique already used by many.

7) Improve Material and methods description with the mentioned lacking information.

*Reviewer #1 (Recommendations for the authors):*

The study is well performed and the experimental design has no specific flaws. However, the conceptual framework should be modified starting from the title and the introduction.

1) The analysis is based on already differentiated cells. Many of the TF involved in the specification may no longer be expressed. This needs to be acknowledged and discussed. Indeed, the authors have not been able to identify additional TFs involved in cell specification, besides those that were already known to have such a function. Even the analysis of tbx2a derives from that of tbx2b, which was in part known. This does not rest value on the analysis and the findings. However, it would be more realistic to state that the analysis offers a list of TF that are likely involved in photoreceptor homeostasis. This should be acknowledged and properly discussed, resting emphasis on cell specification.

2) The data obtained with the F0 screen do not support the idea that tbx2 is a "master regulator" of photoreceptor fate. In my opinion, defining a gene as a "master regulator" is always questionable, but, in this case, there is no evidence for such a definition. If any, I would define tbx2 as a terminal selector.

3) The discussion related to the F0 screen is too long given that the techniques have been used in several studies in addition to the two referenced in the manuscript. I suggest shortening this part of the discussion and giving it the proper importance.

4) As already mentioned, the introduction should be adapted to the experimental design based on the use of already specified and terminally differentiated cells. Similar considerations apply to the discussion.

5) The section of the M and M related to the RNA sample collection should be improved. It was somewhat hard to understand that each sample was composed of 20 cells and the number of samples that have been analyzed for each photoreceptor subtype. In other words, the number of biological replicas should be clearly stated.

*Reviewer #2 (Recommendations for the authors):*

The methods are sound, the results are presented in a well-organized fashion, and the discussion is largely appropriate however several weaknesses were identified.

The Mann-Whitney test was applied to the analysis of cell counts. Unfortunately, this test is not appropriate for multiple comparisons in Figure 4D, 5C, 6C, Suppl Figure 4. The authors may want to consider a test similar to Kruskal-Wallis. The Mann-Whitney test assumes the sample came from a single population so is sensitive to large differences in variance. Mann-Whitney can give an interpretation of significant results when the distributions of the two samples are very different.

For teleosts, more appropriate to use RH2 or the opn1mw1# paralogs of Green Opsin, not an M opsin. Once decided, use the same terminology for the other opsin.

The claim that tbx2 is a master regulatory gene of photoreceptor development is not supported by the data. Master regulators by definition are genes that occupy the very top of a regulatory hierarchy. No data is presented to demonstrate how tbx2 acts. Considering the data, tbx2 appears to regulate UV vs rod fates but only opsin expression in blue and red cones, not their fate. Master regulator is a high bar.

Secondly, the use of a novel term like two-factor-authentication is unnecessary and not supported by the data. The use of a novel term suggests other avenues have been exhausted. But any one of a number of simple to more complicated, known genetic interactions could be at play. The limitation of the study is that the phenotypes are solely based upon analysis of F0 larvae (G0 may be more appropriate as F = filia) which is not amendable to tests of genetic interaction to draw a firm conclusion.

*Reviewer #3 (Recommendations for the authors):*

p4, L216-219, "When ranked by average expression levels across all samples, neurod1 was revealed as …": Cite Figure 2A.

p7, Figure 5 legend requires its title.

p8, L441: "Figure 3 —figure supplement 1" > "Figure 4 —figure supplement 1".

p10, L636: "Ø[nr2e3]" > "FØ[nr2e3]".

p12, L844: Add a citation(?) in the parentheses.

P21, Figure 1—figure supplement 3: There are many citations of "Angueyra et al. 2021" in the panels and legend of this figure without its corresponding in the reference list. Does it mean this manuscript? Also, the figure legend cited "Ogawa et al. 2021" twice, but it should be "Ogawa and Corbo, 2021".

[Editors' note: further revisions were suggested prior to acceptance, as described below.]

Thank you for resubmitting your work entitled "Transcription factors underlying photoreceptor diversity" for further consideration by *eLife*. Your revised article has been evaluated by Marianne Bronner (Senior Editor) and a Reviewing Editor.

The manuscript has been improved but there are some remaining issues that need to be addressed, as outlined below:

Please also pay attention to the use of the term identity as this is used somewhat superficially. Cones types cannot be defined only on the basis of an opsin expression.

*Reviewer #1 (Recommendations for the authors):*

The present manuscript offers valuable transcriptomic data sets of manually picked adult zebrafish photoreceptors from dissociated retinas of different transgenic lines, in which rods and cones (UV, S, L, M) were marked by the fluorescent reporter proteins. This is a very valuable approach because it allows the selection of pure and healthy cells. After an initial global analysis, the authors focus on transcription factors that are differentially expressed in the five photoreceptors cell types that they analyze, identifying a large number of them with still unidentified functions. This is very valuable information that can be exploited to understand photoreceptor cell types specificity. The authors further investigate the activity of the two tbx2 zebrafish paralogues, showing that both paralogs are required for the generation of UV cones as well as for establishing and maintaining the L and S cones' identity. Overall this is an interesting and well-performed study with information that can be valuable for a large number of readers.

The authors have addressed the large majority of the reviewers' comments, with an extensive revision of the manuscript that now offers a better conceptual framework for the study.

*Reviewer #2 (Recommendations for the authors):*

The authors adequately responded to all major concerns.

*Reviewer #3 (Recommendations for the authors):*

The manuscript was largely improved by this revision. Most of the points I previously raised in this part were appropriately revised, except for Figure 5. In the revised manuscript, this figure lacks not only the title but also its whole legend, and should be correctly revised.

---

## [Author Response]

Essential revisions:The following points raised by the reviewers need to be addressed.1) The analysis is based on already differentiated cells and therefore many of the TF needed for photoreceptors specification might not be expressed. This needs to be considered in the title, introduction, and discussion.

We have made appropriate changes throughout the manuscript to avoid the potential mislead in the focus of our study. We have clarified that each gene in our list of differentially-expressed transcription factors (TF) is likely to have specific functions in adult photoreceptors, and some of them might also have a role in the generation of subtypes. We acknowledge that “*photoreceptor specification*” should perhaps be a protected term in our field. For this reason, we have attempted to clarify that photoreceptor development has many stages, and that the initial focus of our screening was on the *generation of photoreceptor subtypes*, and not on earlier stages that lead to the broad specification of photoreceptors as a distinct cell class. In the retina, and other areas of the nervous system, the generation of cell subtypes is usually a later stage of development, and the activation and repression of genetic programs that lead to specific subtype identities are important throughout the lifespan of that neuron, requiring continual expression of specific transcription factors (Deneris and Hobert, 2014).

To address the expression of our candidate genes during photoreceptor development, we used the RNA-seq dataset from Hoang *et al.*,2020, which includes data of embryonic and larval retinal cells, to evaluate the level of expression of our identified transcription factors at the different stages of photoreceptor development. Our initial findings had shown that many of the TF expressed by adult photoreceptors correspond to genes known to be involved in photoreceptor development (blue bars in Figure 2A). In this new analysis, we find that most of the TF that we identify as differentially expressed between photoreceptor subtypes in adults are also expressed in retinal progenitors, photoreceptor progenitors and/or developing photoreceptors, and, for the most part, at higher levels than in adult stages. This new analysis is now included as Figure 2 – Supplement 2. This highlights the ability to detect genes expressed at low levels by photoreceptors in our dataset, lends support to our selection of candidate genes, and reinforces the utility of our screening approach to sort through this long list of candidates. For example, *skor1a* was by all markers a prime candidate: (1) its expression is very specific to UV and S cones; (2) it is known to interact with MEIS1 and other genes involved in photoreceptor development; (3) its expression is highest in the early stages of cone development, mirroring *foxq2*. The negative results presented in our screen provide useful information for our understanding of subtype generation and the players that are actually involved in this process. Currently, we have sparse information on further candidates; additional screening, using the methods and tools presented here, will be invaluable to directly test hypotheses and direct future studies. The detailed code to reanalyze the developmental dataset has been made openly available in our website (./content/code/R_SeuratAnalysis/Hoang2020_10x_photoreceptors/), and we have added plotting routines in our interactive browser to replicate panels in this new figure and to explore the developmental expression changes of any other gene (./content/dr_photoPlotter_Dev.ipynb).

2) Revise statistical analysis using different statistical tests such as Kruskal-Wallis or similar.

We have reanalyzed our data using Kruskal-Wallis tests. In comparisons that involved multiple groups and showed statistical significance, we used Conover-Iman *posthoc* tests between pairs and used Bonferroni correction of p-values. We have made the appropriate changes in figure legends and manuscript. Our conclusions based on significance remain largely unchanged.

3) Revise the claim that tbx2 is a master regulatory gene of photoreceptor development because not supported by the data and is rather inappropriate.

We have removed this claim and replaced it with “Tbx2 plays multiple roles in the generation and maintenance of photoreceptor subtypes”.

4) Revise inconsistent results related to Figure 1-supplement 1.

In Figure 1 – Supplement 1, we had attempted to provide a summarized figure of all phototransduction genes, but the big differences in expression levels — in particular, the high expression of opsins genes — forced us to use gene-by-gene normalization for display. Without normalization, the expression of *opn1mw4* is very low across all samples, and its detection in that sole S-cone sample can likely be attributed to some degree of inherent noise in our methods. We have revised Figure 1 – Supplement 1: we find that we can avoid gene-by-gene normalization and still provide a good summary of the expression of phototransduction genes if the heatmap is broken down by gene families, which have more similar expression levels. In addition, we have added caveats to the use of the Tg(*opn1mw2*:*egfp*) line as our sole M-cone marker in the Results section describing our RNA-seq approach, including our inability to provide data on Opn1mw4-expressing M cones.

5) Revise nomenclature for M opsin, FØ and the use of terms like two-factor-authentication.

As recommended, we have changed all instances to F0. We have also removed the term “two-factor-authentication” from our Discussion. For RNA-seq analysis and gene nomenclature, our study uses the latest genome annotation for zebrafish (*GRCz11*), in which the historical nomenclature of *rh2.1-4* has been deprecated and replaced by *opn1mw1-4*. We have avoided naming these proteins (or cones) “Green Opsins” (or green cones) as their peak sensitivity is not restricted to the wavelengths that humans perceive as “green.” Although still imperfect, we have adopted the use of UV, S, M and L cones to refer to photoreceptor subtypes, and thus referred to opsins in the same way. To avoid any confusion, we have made clarifications in the Introduction and Results section to define these terms unambiguously, and as part of the labels in Figure 1F.

6) Revise the discussion related to the F0 screening, which is rather long for a technique already used by many.

We made revisions to address this issue, removing general details about F0 screening in both Results and Discussion.

7) Improve Material and methods description with the mentioned lacking information.

We have revised the Materials and methods section to include:

– Description of time of day for sample collection for RNA-seq

– Clear statement of number of biological replicates

– Changes in our statistical tests

Reviewer #1 (Recommendations for the authors):The study is well performed and the experimental design has no specific flaws. However, the conceptual framework should be modified starting from the title and the introduction.1) The analysis is based on already differentiated cells. Many of the TF involved in the specification may no longer be expressed. This needs to be acknowledged and discussed. Indeed, the authors have not been able to identify additional TFs involved in cell specification, besides those that were already known to have such a function. Even the analysis of tbx2a derives from that of tbx2b, which was in part known. This does not rest value on the analysis and the findings. However, it would be more realistic to state that the analysis offers a list of TF that are likely involved in photoreceptor homeostasis. This should be acknowledged and properly discussed, resting emphasis on cell specification.

As stated in Essential Revision #1, we have made changes throughout the manuscript to acknowledge this fact, and to rest emphasis on photoreceptor specification. We have also used existing RNA-seq datasets to probe expression of our candidate genes during development. We find that expression of the targets of our F0 screen is present during eye development (new Figure 2 —figure supplement 2G).

2) The data obtained with the F0 screen do not support the idea that tbx2 is a "master regulator" of photoreceptor fate. In my opinion, defining a gene as a "master regulator" is always questionable, but, in this case, there is no evidence for such a definition. If any, I would define tbx2 as a terminal selector.

We have corrected this claim. In our future studies, using germline *tbx2* mutants, we will try to address if these genes pass all the requirements to be defined as terminal selectors.

3) The discussion related to the F0 screen is too long given that the techniques have been used in several studies in addition to the two referenced in the manuscript. I suggest shortening this part of the discussion and giving it the proper importance.

We have now shortened paragraphs in both the Results and Discussion pertaining to generalities about F0 screens.

4) As already mentioned, the introduction should be adapted to the experimental design based on the use of already specified and terminally differentiated cells. Similar considerations apply to the discussion.

We have revised the introduction, title, and discussion to address this issue.

5) The section of the M and M related to the RNA sample collection should be improved. It was somewhat hard to understand that each sample was composed of 20 cells and the number of samples that have been analyzed for each photoreceptor subtype. In other words, the number of biological replicas should be clearly stated.

We originally attempted to make this information accessible with the diagram in Figure 1C, which explains our sample collection and strategy of pooling 20 cells per retina. In Figure 1D (and all subsequent RNA-seq-related figures), data for each pooled sample has been presented without averaging so that variability in the dataset can be directly assessed. Our Results section also stated this: “For each sample, we collected pools of 20 photoreceptors of a single subtype derived from a single adult retina. After collection, we isolated mRNA and generated cDNA libraries for sequencing using SMART-seq2 technology (Figure 1C). In total, we acquired 6 rod samples and 5 UV-, 6 S-, 7 M- and 6 L-cone samples.”

We realize now that we failed to make this plain in the Materials and methods section and have revised it.

Reviewer #2 (Recommendations for the authors):The methods are sound, the results are presented in a well-organized fashion, and the discussion is largely appropriate however several weaknesses were identified.The Mann-Whitney test was applied to the analysis of cell counts. Unfortunately, this test is not appropriate for multiple comparisons in Figure 4D, 5C, 6C, Suppl Figure 4. The authors may want to consider a test similar to Kruskal-Wallis. The Mann-Whitney test assumes the sample came from a single population so is sensitive to large differences in variance. Mann-Whitney can give an interpretation of significant results when the distributions of the two samples are very different.

We thank our reviewer for this recommendation. We have reanalyzed our data using Kruskal-Wallis tests, paired with Conover-Iman *posthoc* comparisons where appropriate, and made changes in methods, results, figures, and legends. All our main conclusions hold, including statistical significance in:

– loss of rods and UV cones in *nr2e3* F0 mutants.

– loss of S cones and gain of M cones in *foxq2* F0 mutants.

– loss of UV cones and gain of rods in *tbx2a* and *tbx2b* F0 mutants.

– misexpression of GFP in L cones in *tbx2a* F0 mutants and in S cones in *tbx2b* F0 mutants.

– main differences in the qPCR analysis.

For teleosts, more appropriate to use RH2 or the opn1mw1# paralogs of Green Opsin, not an M opsin. Once decided, use the same terminology for the other opsin.

See Essential revision #5

The claim that tbx2 is a master regulatory gene of photoreceptor development is not supported by the data. Master regulators by definition are genes that occupy the very top of a regulatory hierarchy. No data is presented to demonstrate how tbx2 acts. Considering the data, tbx2 appears to regulate UV vs rod fates but only opsin expression in blue and red cones, not their fate. Master regulator is a high bar.

We have removed this claim and thank the reviewer for this clarification.

Secondly, the use of a novel term like two-factor-authentication is unnecessary and not supported by the data. The use of a novel term suggests other avenues have been exhausted. But any one of a number of simple to more complicated, known genetic interactions could be at play. The limitation of the study is that the phenotypes are solely based upon analysis of F0 larvae (G0 may be more appropriate as F = filia) which is not amendable to tests of genetic interaction to draw a firm conclusion.

We have also removed this term and agree that F0-screening is a just first step towards the study of positive hits and further studies are warranted.

Reviewer #3 (Recommendations for the authors):p4, L216-219, "When ranked by average expression levels across all samples, neurod1 was revealed as …": Cite Figure 2A.

We have included a reference to Figure 2A.

p7, Figure 5 legend requires its title.

We have included this legend; this was due to an error during the compilation of the pdf that is now corrected. We apologize for the omission.

p8, L441: "Figure 3 —figure supplement 1" > "Figure 4 —figure supplement 1".

We have corrected this mistake.

p10, L636: "Ø[nr2e3]" > "FØ[nr2e3]".

Corrected.

p12, L844: Add a citation(?) in the parentheses.

This has been corrected to correctly display the web address to our interactive plotter.

P21, Figure 1—figure supplement 3: There are many citations of "Angueyra et al. 2021" in the panels and legend of this figure without its corresponding in the reference list. Does it mean this manuscript? Also, the figure legend cited "Ogawa et al. 2021" twice, but it should be "Ogawa and Corbo, 2021".

We have changed this citation to “Current study” to avoid confusion and corrected the other citation.

References

Biehlmaier, O., S. C. Neuhauss, and K. Kohler. 2001. “Onset and Time Course of Apoptosis in the Developing Zebrafish Retina.” *Cell and Tissue Research* 306 (2): 199–207. https://doi.org/10.1007/s004410100447.

Blume, Zachary I., Jared M. Lambert, Anna G. Lovel, and Diana M. Mitchell. 2020. “Microglia in the Developing Retina Couple Phagocytosis with the Progression of Apoptosis via P2ry12 Signaling.” Developmental Dynamics : An Official Publication of the American Association of Anatomists 249 (6): 723–40. https://doi.org/10.1002/dvdy.163.

Deneris, Evan S., and Olivier Hobert. 2014. “Maintenance of Postmitotic Neuronal Cell Identity.” *Nature Neuroscience* 17 (7). https://doi.org/10.1038/nn.3731.

Lusk, Sarah, and Kristen M. Kwan. 2022. “Pax2a, but Not Pax2b, Influences Cell Survival and Periocular Mesenchyme Localization to Facilitate Zebrafish Optic Fissure Closure.” Developmental Dynamics: An Official Publication of the American Association of Anatomists 251 (4): 625–44. https://doi.org/10.1002/dvdy.422.

[Editors' note: further revisions were suggested prior to acceptance, as described below.]

The manuscript has been improved but there are some remaining issues that need to be addressed, as outlined below:Please also pay attention to the use of the term identity as this is used somewhat superficially. Cones types cannot be defined only on the basis of an opsin expression.

We have made changes to the manuscript to limit the use (and potential misuse) of the term identity. We also discuss the need to examine molecular markers beyond opsins to define photoreceptor identity.

Reviewer #3 (Recommendations for the authors):The manuscript was largely improved by this revision. Most of the points I previously raised in this part were appropriately revised, except for Figure 5. In the revised manuscript, this figure lacks not only the title but also its whole legend, and should be correctly revised.

On our end, Figure 5 and its legend are present. We will ensure this is the case in the new submission but can provide files by other means, if necessary. We also removed any previous versions of the manuscript from the submission platform to avoid any confusion.